# PaAno: Patch-Based Representation Learning for Time-Series Anomaly Detection

**Jinju Park, Seokho Kang**[*]
Department of Industrial Engineering, Sungkyunkwan University
`{apfhsk777, s.kang}@skku.edu`

## Abstract

Although recent studies on time-series anomaly detection have increasingly adopted ever-larger neural network architectures such as Transformers and Foundation models, they incur high computational costs and memory usage, making them impractical for real-time and resource-constrained scenarios. Moreover, they often fail to demonstrate significant performance gains over simpler methods under rigorous evaluation protocols. In this study, we propose **Pa**tch-based representation learning for time-series **Ano**maly detection (**PaAno**), a lightweight yet effective method for fast and efficient time-series anomaly detection. PaAno extracts short temporal patches from time-series training data and uses a 1D convolutional neural network to embed each patch into a vector representation. The model is trained using a combination of triplet loss and pretext loss to ensure the embeddings capture informative temporal patterns from input patches. During inference, the anomaly score at each time step is computed by comparing the embeddings of its surrounding patches to those of normal patches extracted from the training time-series. Evaluated on the TSB-AD benchmark, PaAno achieved state-of-the-art performance, significantly outperforming existing methods, including those based on heavy architectures, on both univariate and multivariate time-series anomaly detection across various range-wise and point-wise performance measures. The source code is available at `https://github.com/jinnnju/PaAno`.

## 1 Introduction

Time-series data, a collection of temporally ordered observations, are pervasive across a wide range of domains, including industrial sensor measurements, financial market transactions, and healthcare monitoring (Yue et al., 2022; Jia et al., 2024). A defining characteristic of time-series data is the presence of temporal dependencies among observations, shaped by the temporal context and underlying system dynamics (Lai et al., 2018; Leung et al., 2023; Islam, 2024). However, these dependencies can be disrupted by various factors such as system faults, external disturbances, or human errors. Such anomalies may manifest as sudden spikes, abrupt drops, or sustained deviations in certain observations, as illustrated in Figure 1. *Time-Series Anomaly Detection* aims to identify time points or segments within a sequence whose patterns deviate significantly from expected normal behavior (Paparrizos et al., 2022; Zhou et al., 2023a; Sarfraz et al., 2024). Because these anomalies are often indicative of underlying issues, accurate and timely detection is crucial for ensuring reliability and safety in real-world applications.

Recent studies on time-series anomaly detection have introduced large-scale neural network architectures, such as Transformers (Xu et al., 2021; Tuli et al., 2022; Wu et al., 2022; Yue et al., 2024) and Foundation models (Rasul et al., 2023; Zhou et al., 2023b; Goswami et al., 2024), aiming to capture long-term temporal dependencies and cross-variable relationships effectively. Nevertheless, Sarfraz et al. (2024) and Liu & Paparrizos (2024) have highlighted an illusion of progress in their methods by revealing limitations in existing evaluation practices, including structural flaws in benchmark datasets and inadequate performance measures that rely on point adjustment or threshold tuning. When evaluated under rigorous protocols designed to mitigate these issues, these sophisticated

---
[*]Corresponding author.

large-scale neural network-based methods did not demonstrate significant advantages over simpler methods (Sarfraz et al., 2024; Liu & Paparrizos, 2024). Moreover, the high computational cost and memory usage further constrain their practicality in real-time or resource-constrained scenarios.

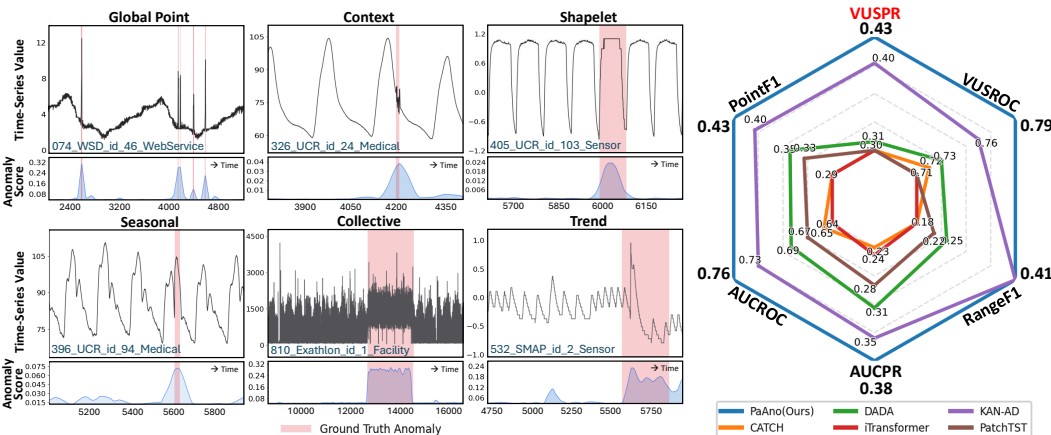

Figure 1: Illustrative results of PaAno, demonstrating strong capability in detecting diverse types of time-series anomalies. Datasets from TSB-AD-U (Liu & Paparrizos, 2024).

Figure 2: Anomaly detection performance of PaAno and recent methods on TSB-AD-M.

In this work, we propose **Pa**tch-based representation learning for time-series **Ano**maly detection (**PaAno**), a lightweight time-series anomaly detection method based on patch-based representation learning. While recent studies on time-series anomaly detection have often relied on forecasting-based or reconstruction-based methods, representation-based methods remain relatively underexplored and only a limited subset of recent time-series representation learning studies has explicitly targeted anomaly detection (Zamanzadeh Darban et al., 2024). PaAno addresses this gap by introducing a patch-level embedding space tailored to capturing subtle deviations in time-series data while remaining invariant to small temporal shifts. Given a time-series training dataset consisting only of normal patterns, PaAno extracts short temporal segments, called *Patches*, by shifting a window one time step at a time. As the representation model, a compact 1D Convolutional Neural Network (1D-CNN) is used to embed these patches into vector representations. The model is trained with a learning objective that integrates metric learning and a self-supervised pretext task, encouraging the embedding vectors to be informative and discriminative with respect to temporal patterns in the input patches. After training, it constructs a memory bank as a set of embedding vectors representing core patches from the training dataset. During inference, the anomaly score at each time step is computed based on the distances between the embeddings of the surrounding patches and their closest embeddings in the memory bank. Since PaAno does not rely on large-scale or heavily tuned architectures, it is fast and efficient, and well-suited for real-time anomaly detection in resource-constrained environments. We evaluated the effectiveness of PaAno on the TSB-AD benchmark (Liu & Paparrizos, 2024) with performance measures that avoid point adjustment and threshold tuning to ensure accurate evaluation. PaAno consistently outperformed existing methods across all measures.

Our main contributions are summarized as follows:

- PaAno introduces a novel representation-based framework that constructs a discriminative patch-level embedding space tailored for time-series anomaly detection.

- PaAno uses a lightweight 1D-CNN model, enabling fast and efficient time-series anomaly detection compared to recent methods that rely on heavy neural network architectures.

- PaAno consistently achieves state-of-the-art results on both univariate and multivariate time-series anomaly detection tasks across both range-wise and point-wise performance measures.

- PaAno shows high robustness to hyperparameter configurations, indicating that it does not require extensive hyperparameter tuning.

## 2 RELATED WORK

### 2.1 TIME-SERIES ANOMALY DETECTION

Time-series anomaly detection is formulated as learning from a time-series training dataset $\mathbf{X} = (\mathbf{x}_1, \ldots, \mathbf{x}_N)$ consisting of $N$ sequential observations, where $\mathbf{x}_t \in \mathbb{R}^d$ represents the observation at time step $t$, to predict whether a query observation $\mathbf{x}_{t_*}$ is anomalous. If each observation contains only a single variable (*i.e.*, $d = 1$), the task is referred to as univariate time-series anomaly detection. If there is more than one variable (*i.e.*, $d > 1$), it is called multivariate time-series anomaly detection, where detection relies on the dependencies among multiple variables across time steps. Time-series anomaly detection is typically categorized into three paradigms based on the availability of labels in the training dataset (Choi et al., 2021; Boniol et al., 2024; Zamanzadeh Darban et al., 2024): unsupervised, semi-supervised, and supervised. In the unsupervised setting, the training dataset is unlabeled, and there is no explicit distinction between normal and anomalous observations. In the semi-supervised setting, only normal observations are present in the training dataset. In the supervised setting, the training dataset contains both normal and anomalous observations.

In this study, we focus on semi-supervised time-series anomaly detection, which is widely regarded as practical for real-world applications where labeled anomalies are extremely scarce and costly to obtain. Time-series training dataset $\mathbf{X} = (\mathbf{x}_1, \ldots, \mathbf{x}_N)$ is assumed to consist solely of normal observations. The objective is to learn a model from the training dataset $\mathbf{X}$ to detect anomalies in query observations $\mathbf{x}_{t_*}$. At each time step $t_*$, the model produces an anomaly score $s_{t_*} \in \mathbb{R}$, with a higher score indicating a greater likelihood that the corresponding observation $\mathbf{x}_{t_*}$ is anomalous.

### 2.2 TIME-SERIES ANOMALY DETECTION METHODS

A wide range of methods have been proposed for time-series anomaly detection, spanning from classical statistical techniques to modern deep learning architectures (Liu & Paparrizos, 2024). We categorize existing methods into three groups based on the model architecture used: statistical and machine learning, neural network-based, and Transformer-based methods. Our proposed method PaAno belongs to the category of neural network-based methods. PaAno enables fast and efficient anomaly detection by utilizing a lightweight 1D-CNN.

**Statistical and Machine Learning Methods**  Methods in this category use statistical assumption or traditional machine learning algorithms to detect anomalies. Most of these methods were originally designed for point-wise anomaly detection in non-time-series data. They can be further categorized into four sub-groups. First, density-based methods detect points or segments that lie in low-density regions or deviate from estimated underlying distributions, including COPOD (Li et al., 2020), LOF (Breunig et al., 2000), KNN (Ramaswamy et al., 2000), and Matrix Profile (Yeh et al., 2016). Second, boundary-based methods find decision boundaries that separate normal and anomalous data, including OCSVM (Schölkopf et al., 1999), IForest (Liu et al., 2008), and EIF (Hariri et al., 2019). Third, reconstruction-based methods reconstruct the expected normal data and detect anomalies via residual errors. PCA (Aggarwal, 2017), RobustPCA (Paffenroth et al., 2018), and SR (Ren et al., 2019) employ dimensionality reduction. DLinear and NLinear (Zeng et al., 2023) use trend–remainder decomposition with a temporal linear layer for reconstruction. Fourth, clustering-based methods detect anomalies as deviations from learned cluster structures, including KMeansAD (Yairi et al., 2001), CBLOF (He et al., 2003), KShapeAD (Paparrizos & Gravano, 2015; 2017), Series2Graph (Boniol & Palpanas, 2020), and SAND (Boniol et al., 2021).

Additionally, some of these methods have extensions designed for fixed-length segments and we denoted those with the prefix "(Sub)-" (*e.g.*, (Sub)-PCA, (Sub)-KNN), following the practices in TSB-AD (Liu & Paparrizos, 2024).

**Neural Network-Based Methods**  This category comprises conventional, non-Transformer neural network architectures for time-series anomaly detection. Multi-Layer Perceptron (MLP)-based methods detect anomalies through reconstruction errors, including AutoEncoder (Sakurada & Yairi, 2014), USAD (Audibert et al., 2020), and Donut (Xu et al., 2018). Recurrent neural network (RNN)-based methods explicitly model sequences in time-series data to capture temporal dependencies, including LSTMAD (Malhotra et al., 2015) and OmniAnomaly (Su et al., 2019). CNN-based meth-

ods utilize convolutional architectures to extract temporal features from time series, including Deep-AnT (Munir et al., 2018) and TimesNet (Wu et al., 2022). FITS (Xu et al., 2023) processes time series by interpolation in the complex frequency domain. DADA (Shentu et al., 2025) learns robust representations through adaptive information bottlenecks and dual adversarial decoders, enabling zero-shot anomaly detection across multi-domain time-series. KAN-AD (Zhou et al., 2025) reformulates time-series modeling via Fourier enhanced Kolmogorov–Arnold networks, achieving highly smooth normal pattern approximation with minimal parameters for efficient anomaly detection.

**Transformer-Based Methods** Transformer architectures have been increasingly adopted to better capture long-range dependencies in time-series data. AnomalyTransformer (Xu et al., 2021) and TranAD (Tuli et al., 2022) compute anomaly scores based on attention discrepancies. DCdetector (Yang et al., 2023) employs dual attention with contrastive learning and CATCH (Wu et al., 2025) applies frequency patching with a channel fusion module. PatchTST (Nie et al., 2023) tokenizes subseries-level patches with channel-independent weights. iTransformer (Liu et al., 2024) adopts an inverted view that swaps time and variable dimensions.

Foundation models, pretrained on large-scale time-series data, are adopted in time-series anomaly detection to enable zero-shot and few-shot detection capabilities. Chronos (Ansari et al., 2024) and MOMENT (Goswami et al., 2024) based on T5-style encoder-decoder architectures. Lag-Llama (Rasul et al., 2023), TimesFM (Das et al., 2024), and OFA (Zhou et al., 2023b) utilize decoder-only Transformer architectures.

## 2.3 Issues in Evaluation Practices

Recent studies on time-series anomaly detection have often adopted evaluation protocols that introduce systematic biases, undermining the validity of reported results. First, several commonly used benchmark datasets suffer from structural flaws (Liu & Paparrizos, 2024). These include labeling inconsistencies, where some anomaly-labeled observations are indistinguishable from normal patterns, and unrealistic assumptions about anomaly distributions, such as restricting anomalies to appear only once or at the end of a time series. Second, the conventional use of performance measures that rely on point adjustment and threshold tuning can misleadingly inflate scores and hinder fair comparison across methods (Sarfraz et al., 2024; Bhattacharya et al., 2024; Paparrizos et al., 2022; Liu & Paparrizos, 2024).

To address these issues, we adopt the TSB-AD benchmark (Liu & Paparrizos, 2024), which mitigates dataset-related flaws by correcting labeling inconsistencies and modeling anomalies under more realistic assumptions. In addition, we completely remove point adjustment from the evaluation protocol and include four threshold-independent measures, thereby eliminating biases from miscellaneous convention. Further discussions of evaluation measures are provided in Appendix C.

# 3 Proposed Method

## 3.1 Overview

**Local Temporal Dependencies** Recent Transformer models have strengths in modeling long-term temporal dependencies by processing long time-series jointly (Zamanzadeh Darban et al., 2024; Liu et al., 2024; Wu et al., 2025). However, anomaly detection in time series often relies on localized patterns within short intervals. Time-series are typically strongly correlated with their immediate neighbors but only weakly related to distant points, with this tendency particularly pronounced around anomalies (Xu et al., 2021; Yue et al., 2024). Using Transformers with global self-attention for modeling long sequences can dilute local temporal dependencies, as the mechanism is inherently locality-agnostic, leading to insensitivity to local context (Li et al., 2019; 2023; Oliveira & Ramos, 2024). Consequently, these models often struggle to capture the fine-grained local dynamics in shorter subsequences that are crucial for time-series anomaly detection, particularly for detecting point or contextual anomalies.

**Patch-Based Formulation** To enable accurate and efficient anomaly detection, we introduce an inductive bias toward locality in PaAno. Our method PaAno is motivated by recent advances in visual anomaly detection that leverage patch-based representation learning (Defard et al., 2021; Yi &

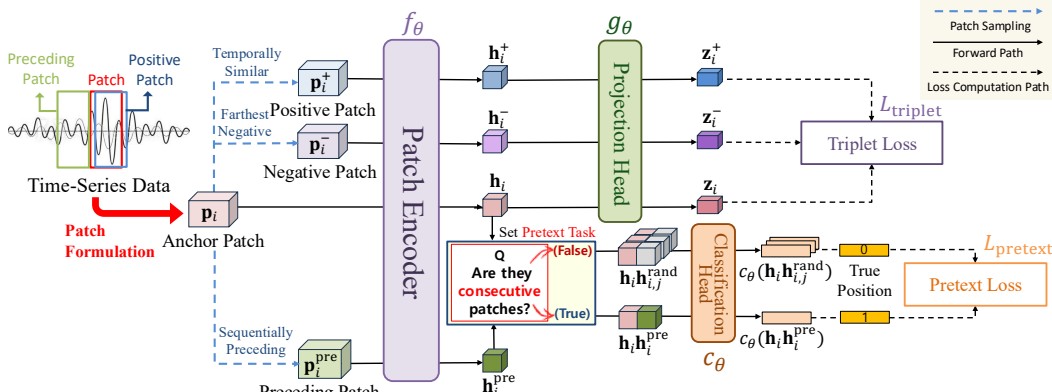

Figure 3: Training procedure of PaAno. The training dataset is split into patches. Using the patch set, three model components—a patch encoder, a projection head, and a classification head—are trained with the training objective that consists of two losses. Triplet loss encourages temporally similar patches to have closer embeddings in the projected space, and pretext loss guides the patch encoder to learn temporal relationships by predicting whether two patches are consecutive.

Yoon, 2020; Roth et al., 2022; Yoon et al., 2023). These methods have shown superior performance on benchmark datasets such as MVTec-AD (Bergmann et al., 2019), where normal images within the same class exhibit highly consistent and repetitive spatial patterns, while anomalies disrupt these regularities.

We observe that time-series often exhibit analogous characteristics. Normal time-series display repetitive temporal patterns and strong local dependencies, whereas anomalies typically break such short-range regularities. PaAno aims to precisely capture these temporal dynamics through patch-based representation learning. Given a training sequence $\mathbf{X}$, we extract overlapping fixed-length subsequences, referred to as patches, of window size $w$ using a unit-stride sliding window, yielding $\mathcal{P} = \{\mathbf{p}_t\}_{t=1}^{N-w+1}$ where $\mathbf{p}_t = (\mathbf{x}_t, \ldots, \mathbf{x}_{t+w-1})$. Patches serve as the fundamental units for anomaly detection. For each patch, we apply instance normalization (Kim et al., 2022b; Yang et al., 2023; Wu et al., 2025), which standardizes all channels within the patch to zero mean and unit variance. Reducing patch-level variability in mean and variance improves the stability of patch-level representations and increases robustness to distributional shifts such as regime changes or drift. To enrich patch embeddings, PaAno leverages the sequential continuity and temporal similarity across patches, which enable effective learning of local temporal dependencies essential for anomaly detection.

Figure 3 provides a schematic overview of PaAno, including its model architecture and training objective. The pseudocode of the training procedure is presented in Appendix A.

## 3.2 MODEL ARCHITECTURE

The model architecture of PaAno comprises three main components: a patch encoder $f_\theta$, a projection head $g_\theta$, and a classification head $c_\theta$. The patch encoder $f_\theta$ is a 1D-CNN that embeds temporal patterns from the input patch $\mathbf{p} \in \mathbb{R}^{w \times d}$ into a vector representation $\mathbf{h} \in \mathbb{R}^l$. This patch-level embedding $\mathbf{h}$ serves as the input to both the projection head $g_\theta$ and classification head $c_\theta$. The projection head $g_\theta$ is an MLP that transforms $\mathbf{h}$ into its projection $\mathbf{z}$. This projected embedding is used for metric learning, encouraging the encoder $f_\theta$ to extract features that are discriminative with respect to the temporal patterns in the input patch. The classification head $c_\theta$ is another MLP that takes the embeddings of two patches as inputs to predict whether they are temporally consecutive, thereby encouraging the encoder $f_\theta$ to capture sequential relationships among patches. Once the model is trained, only the patch encoder $f_\theta$ is retained for anomaly detection in future observations.

### 3.3 TRAINING OBJECTIVE

Given a minibatch $\mathcal{B} = \{\mathbf{p}_i\}_{i=1}^M$ sampled from the patch set $\mathcal{P}$ at each training iteration, the training objective for patch-based representation learning combines two loss functions: a triplet loss $\mathcal{L}_{\text{triplet}}$ and a pretext loss $\mathcal{L}_{\text{pretext}}$. The overall training objective is given by:

$$\mathcal{L} = \mathcal{L}_{\text{triplet}} + \lambda \cdot \mathcal{L}_{\text{pretext}}, \tag{1}$$

where the weight $\lambda$ controls the contribution of the pretext loss $\mathcal{L}_{\text{pretext}}$. Details of each loss function are described below.

**Triplet Loss**   The triplet loss is a loss function introduced in deep metric learning to learn an embedding space where an anchor example is closer to a positive example than to a negative one by a specified margin (Schroff et al., 2015). To promote the patch encoder $f_\theta$ to extract embeddings that capture temporal patterns in the input patches, we adopt triplet loss so that patches with similar temporal patterns are embedded close together, while those with dissimilar patterns are pushed apart.

For each patch $\mathbf{p}_i \in \mathcal{B}$ as the anchor, the positive patch $\mathbf{p}_i^+$ is obtained by randomly shifting the anchor $\mathbf{p}_i$ within $r$ time steps, excluding zero shift. This ensures that the anchor and positive patches exhibit similar temporal patterns. We define the negative patch $\mathbf{p}_i^-$ as the farthest negative, chosen as the patch in the minibatch $\mathcal{B}$ that has the largest cosine distance to $\mathbf{p}_i$ in the embedding space, (*i.e.*, $\mathbf{p}_j \in \mathcal{B} \setminus \{\mathbf{p}_i\}$) that maximizes $\text{dist}(f_\theta(\mathbf{p}_i), f_\theta(\mathbf{p}_j))$. After the patches $\mathbf{p}_i$, $\mathbf{p}_i^+$, and $\mathbf{p}_i^-$ are encoded and projected into $\mathbf{z}_i$, $\mathbf{z}_i^+$, and $\mathbf{z}_i^-$ through the encoder $f_\theta$ and projection head $g_\theta$, the loss $\mathcal{L}_{\text{triplet}}$ is computed as:

$$\mathcal{L}_{\text{triplet}} = \frac{1}{M} \sum_{i=1}^M \max\left(0, \text{dist}(\mathbf{z}_i, \mathbf{z}_i^+) - \text{dist}(\mathbf{z}_i, \mathbf{z}_i^-) + \delta\right), \tag{2}$$

where $\delta$ is the margin hyperparameter that denotes the minimum distance the anchor must be closer to the positive than to the negative and $M$ is size of minibatch. Minimizing $\mathcal{L}_{\text{triplet}}$ encourages the anchor to be closer to the positive patch than to the negative patch in the embedding space, thereby making the embedding space robust to small temporal shifts while remaining sensitive to meaningful differences. We expect this to produce well-organized clusters of normal patches, with unseen future anomalous patches lying far from them and thus being effectively identified.

**Pretext Loss**   The pretext loss is inspired by Yi & Yoon (2020)'s study on visual anomaly detection based on patch-based representation learning, where the training objective includes a patch-level classification task to predict the relative positions of image patches. This pretext task enhances patch embeddings by promoting spatial awareness. We adapt this idea to time-series data by formulating a patch-level classification task that predicts whether two patches are temporally consecutive, thereby guiding the patch encoder $f_\theta$ to better capture temporal relationships among patches.

For each anchor patch $\mathbf{p}_i \in \mathcal{B}$, we select the preceding patch $\mathbf{p}_i^{\text{pre}}$ that is exactly $w$ time steps ahead, such that it is temporally preceding to the anchor $\mathbf{p}_i$. We also draw $U$ random patches, $\mathbf{p}_{i,1}^{\text{rand}}, \ldots, \mathbf{p}_{i,U}^{\text{rand}}$ from $\mathcal{B} \setminus \{\mathbf{p}_i\}$. The classification head $c_\theta$ takes a pair of patch embeddings as input and outputs the probability estimate indicating whether the two patches are temporally consecutive. After obtaining the embeddings for $\mathbf{p}_i$, $\mathbf{p}_i^{\text{pre}}$, and $\mathbf{p}_{i,j}^{\text{rand}}$ using the encoder $f_\theta$, the loss $\mathcal{L}_{\text{pretext}}$ is computed as:

$$\mathcal{L}_{\text{pretext}} = \frac{1}{M} \sum_{i=1}^M \left[ -\log c_\theta(\mathbf{h}_i, \mathbf{h}_i^{\text{pre}}) - \frac{1}{U} \sum_{j=1}^U \log\left(1 - c_\theta(\mathbf{h}_i, \mathbf{h}_{i,j}^{\text{rand}})\right) \right]. \tag{3}$$

Minimizing $\mathcal{L}_{\text{pretext}}$ encourages the classification head $c_\theta$ to assign a high probability to a pair consisting of an anchor patch and its preceding patch, and a low probability to a pair consisting of an anchor patch and a random patch. This loss is applied only during the early stage of training to expedite the learning of temporal relationships among patches, thereby stabilizing representation learning when the embedding space is not yet structured.

### 3.4 MEMORY BANK

After training, the patch encoder $f_\theta$ forms a embedding space where similar normal patches are tightly grouped, while distinct normal patterns occupy different regions of the space. A memory

bank $\mathcal{M}$ is constructed as the set of embeddings of patches $\mathbf{p}_t \in \mathcal{P}$ obtained using $f_\theta$:

$$\mathcal{M} = \{f_\theta(\mathbf{p}_t) \mid \mathbf{p}_t \in \mathcal{P}\}. \tag{4}$$

For anomaly detection, it serves as a collection of cohesive clusters of normal patches present in the training dataset $\mathbf{X}$, providing a clear reference for identifying anomalies in future observations.

A practical consideration is that the memory bank $\mathcal{M}$ grows with the size of the training dataset $\mathbf{X}$, leading to increased computational costs and storage requirements for anomaly detection (Yi & Yoon, 2020; Roth et al., 2022). To address this, we apply coreset subsampling to reduce the size of $\mathcal{M}$. We perform $K$-means clustering on $\mathcal{M}$ to derive $K$ clusters. For each $i$-th cluster, we select the vector $\mathbf{m}_i \in \mathcal{M}$ that is closest to its centroid. The resulting $K$ representative vectors, denoted by $\mathbf{m}_1, \ldots, \mathbf{m}_K$, are used to construct a reduced memory bank $\hat{\mathcal{M}}$:

$$\hat{\mathcal{M}} = \{\mathbf{m}_i\}_{i=1}^{K}. \tag{5}$$

This reduction preserves representative coverage of the original memory bank while significantly reducing its size, thereby enhancing the efficiency of anomaly detection.

## 3.5 ANOMALY DETECTION

During inference, the patch encoder $f_\theta$ and reduced memory bank $\hat{\mathcal{M}}$ are used to compute the anomaly score $s_{t_*}$ for a query time step $t_*$.

We first compute patch-level anomaly scores for the patches that include the query time step $t_*$. Let $\mathcal{P}_{t_*} = \{\mathbf{p}_t\}_{t=t_*-w+1}^{t_*}$ denote the set of these patches, where each patch $\mathbf{p}_t = (\mathbf{x}_t, \ldots, \mathbf{x}_{t+w-1})$ is a collection of the $w$ most recent observations starting at time step $t$. Each patch $\mathbf{p}_t \in \mathcal{P}_{t_*}$ is embedded using the encoder $f_\theta$ as $f_\theta(\mathbf{p}_t)$. This embedding

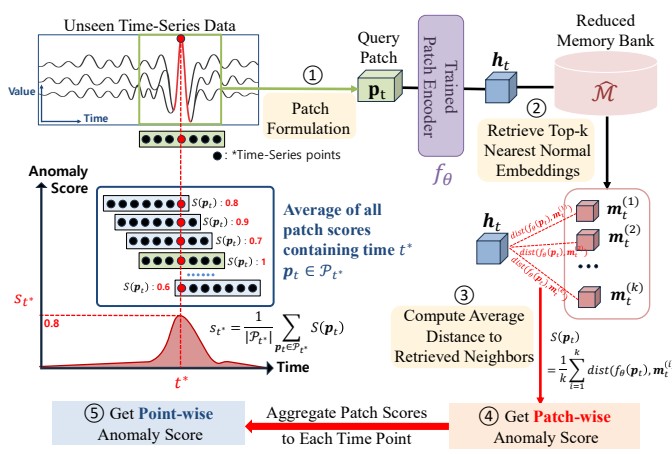

Figure 4: Anomaly detection procedure of PaAno.

is then compared to the vectors in the memory bank $\hat{\mathcal{M}}$ for anomaly scoring. Specifically, the $k$ nearest neighbors in terms of cosine distance are retrieved from $\hat{\mathcal{M}}$, denoted by $\mathbf{m}_t^{(1)}, \ldots, \mathbf{m}_t^{(k)}$. The patch-level anomaly score for $\mathbf{p}_t$, denoted $S(\mathbf{p}_t)$, is computed as:

$$S(\mathbf{p}_t) = \frac{1}{k} \sum_{i=1}^{k} \text{dist}(f_\theta(\mathbf{p}_t), \mathbf{m}_t^{(i)}). \tag{6}$$

This score reflects how dissimilar the patch $\mathbf{p}_t$ is from the normal patterns learned from the training dataset $\mathbf{X}$. For an anomalous patch, its embedding fails to align with any clusters of normal patches in the memory bank, resulting in a high anomaly score.

The final anomaly score for the query time step $t_*$, denoted $s_{t_*}$, is obtained by averaging the patch-level scores of all patches in $\mathcal{P}_{t_*}$:

$$s_{t_*} = \frac{1}{|\mathcal{P}_{t_*}|} \sum_{\mathbf{p}_t \in \mathcal{P}_{t_*}} S(\mathbf{p}_t). \tag{7}$$

The anomaly score $s_{t_*}$ reflects the temporal context across the surrounding patches. A higher value of $s_{t_*}$ means that the behavior around the time step $t_*$ is substantially different from those observed in the training dataset $\mathbf{X}$, and is thus indicative of a potential anomaly. Figure 4 illustrates the anomaly detection procedure of PaAno, and its pseudocode is fully provided in Appendix A.

# 4 EXPERIMENTS

## 4.1 EXPERIMENTAL SETTINGS

**TSB-AD Benchmark**   We used the datasets provided by the TSB-AD benchmark (Liu & Paparrizos, 2024), which were specifically curated to address critical limitations in existing evaluation practices for time-series anomaly detection. The summary statistics are presented in Table 1.

Table 1: Statistics of the TSB-AD benchmark.

| Category | Split | # Time-Series | Avg. Length | Anomaly Ratio |
|---|---|---|---|---|
| TSB-AD-U | Tuning | 48 | 47143.3 | 3.5% |
| | Eval | 530 | 51866.7 | 4.5% |
| TSB-AD-M | Tuning | 20 | 98164.1 | 5.7% |
| | Eval | 180 | 108826.7 | 5.0% |

Figure 1 shows examples of the time-series datasets. The datasets are categorized into two groups based on the number of variables: TSB-AD-U for univariate and TSB-AD-M for multivariate time series. Each group is further divided into a "Tuning" set for hyperparameter optimization and an "Eval" set for performance evaluation. Additionally, each time series has a predefined split point, where the preceding segment is designated for the training dataset.

**Implementation Details**   The patch encoder $f_\theta$ was a 1D-CNN consisting of four 1D convolutional layers followed by global average pooling layer with output size 64. The projection head $g_\theta$ was a two-layer MLP with dimensionality of 256 and the classification head $c_\theta$ was a single-layer MLP. The model was trained for 100 iterations using the AdamW optimizer with a minibatch size $M$ of 512 and a weight decay of 1e−4. The pretext loss weight $\lambda$ was linear decayed from 1 to 0 during the first 20 iterations and fixed at 0 thereafter. The memory bank size was set to $10\%$ of the original patch set $\mathcal{P}$. The number of nearest neighbors $k$ in the anomaly scoring function was set to 3. Each experiment was repeated 10 times with different random seeds, and the average results are reported. Further details of implementation are described in Appendix B.

**Baseline Methods**   The TSB-AD benchmark provides a comprehensive comparison across a total of 40 baseline methods, including 25 statistical and machine learning methods, 8 neural network-based methods, and 7 Transformer-based methods. We additionally include 8 methods for further comparison with recent advances: PatchTST (Nie et al., 2023), DLinear (Zeng et al., 2023), NLinear (Zeng et al., 2023), DCdetector (Yang et al., 2023), iTransformer (Liu et al., 2024), CATCH (Wu et al., 2025), KAN-AD (Zhou et al., 2025), and DADA (Shentu et al., 2025). Among a total of 48 baseline methods, 39 are applicable to univariate time-series anomaly detection and 31 to multivariate time-series anomaly detection. The hyperparameters of all baseline methods were tuned in the same way as for the proposed method. The search spaces used are provided in Appendix B.

**Performance Measures**   For a reliable and comprehensive evaluation, we employed three range-wise and three point-wise measures, respectively. The range-wise measures comprise the VUS-ROC and VUS-PR (Paparrizos et al., 2022; Liu & Paparrizos, 2024; Boniol et al., 2025), together with the Range-wise F1 score (Range-F1). As point-wise measures, we adopt AUC-PR, AUC-ROC, and Point-F1. Following the TSB-AD (Liu & Paparrizos, 2024), VUS-PR was regarded as the primary evaluation measure, while the others were used as complementary measures. Detailed definitions of the measures are presented in Appendix C.

## 4.2 RESULTS AND DISCUSSION

**Anomaly Detection Results**   Tables 2 and 3 summarize the performance of the proposed method and competitive baselines on the univariate (TSB-AD-U) and multivariate (TSB-AD-M) datasets from the TSB-AD benchmark. We report the top-2 performing methods based on VUS-PR, along with those published within the past 4 years (since 2022) for category except for Transformers. The full experimental results and statistical tests for all compared methods are provided in Appendix E.

In univariate time-series anomaly detection (Table 2), PaAno ranked first across all six performance measures, outperforming all baseline methods. The qualitative results in Figure 1 show that PaAno effectively captures diverse types of anomalies, ranging from abrupt point anomalies to contextual anomaly segments. Among the baselines, KAN-AD, which belongs to the neural network–based

Table 2: Experimental results on **univariate time-series anomaly detection** in TSB-AD-U. For each measure, the best and second-best values are indicated in **bold** and underlined. All scores are reported with their rankings as "Score/Rank".

| | Method | VUS-PR | VUS-ROC | Range-F1 | AUC-PR | AUC-ROC | Point-F1 | # Params | Run Time |
|---|---|---|---|---|---|---|---|---|---|
| **Stat&ML** | (Sub)-PCA (2017) | 0.42/2 | 0.76/10 | 0.41/3 | 0.37/3 | 0.71/11 | 0.42/3 | – | 1.5s |
| | KShapeAD (2017) | 0.40/3 | 0.76/10 | 0.40/4 | 0.35/4 | 0.74/5 | 0.39/4 | – | 8.0s |
| | DLinear (2023) | 0.27/20+ | 0.76/10 | 0.24/20+ | 0.23/20+ | 0.67/19 | 0.28/20+ | < 0.1M | 2.9s |
| | NLinear (2023) | 0.26/20+ | 0.74/20+ | 0.21/20+ | 0.20/20+ | 0.63/20+ | 0.24/20+ | < 0.1M | 5.8s |
| **NN** | DeepAnT (2018) | 0.34/13 | 0.79/5 | 0.35/9 | 0.33/5 | 0.71/11 | 0.38/5 | < 0.1M | 2.0s |
| | USAD (2020) | 0.36/10 | 0.71/20+ | 0.40/4 | 0.32/7 | 0.66/20+ | 0.37/8 | < 0.1M | 1.7s |
| | TimesNet (2022) | 0.26/20+ | 0.72/20+ | 0.21/20+ | 0.18/20+ | 0.61/20+ | 0.24/20+ | < 0.1M | 11.2s |
| | FITS (2023) | 0.26/20+ | 0.73/20+ | 0.20/20+ | 0.17/20+ | 0.61/20+ | 0.23/20+ | < 0.1M | 3.1s |
| | DADA (2025) | 0.31/18 | 0.77/8 | 0.31/19 | 0.29/14 | 0.71/11 | 0.33/18 | 1.84M | 0.8s |
| | KAN-AD (2025) | 0.40/3 | 0.82/2 | 0.43/2 | 0.41/2 | 0.80/2 | 0.44/2 | < 0.1M | 12.1s |
| **Transformer** | AnomalyTransformer (2021) | 0.12/20+ | 0.56/20+ | 0.14/20+ | 0.08/20+ | 0.50/20+ | 0.12/20+ | 4.8M | 48.9s |
| | DCdetector (2023) | 0.09/20+ | 0.57/20+ | 0.07/20+ | 0.05/20+ | 0.50/20+ | 0.10/20+ | 0.8M | 5.8s |
| | Lag-Llama (2023) | 0.27/20+ | 0.72/20+ | 0.31/19 | 0.25/20+ | 0.65/20+ | 0.30/20+ | 2.5M | 1220.8s |
| | OFA (2023b) | 0.24/20+ | 0.71/20+ | 0.20/20+ | 0.16/20+ | 0.59/20+ | 0.22/20+ | 81.9M | 171.1s |
| | PatchTST (2023) | 0.30/19 | 0.76/10 | 0.25/20+ | 0.23/20+ | 0.65/20+ | 0.27/20+ | 6.9M | 26.3s |
| | iTransformer (2024) | 0.32/16 | 0.77/8 | 0.22/20+ | 0.21/20+ | 0.67/19 | 0.26/20+ | 6.3M | 9.8s |
| | MOMENT (FT) (2024) | 0.39/5 | 0.76/10 | 0.35/9 | 0.30/12 | 0.69/15 | 0.35/12 | 109.6M | 43.6s |
| | MOMENT (ZS) (2024) | 0.38/8 | 0.75/20 | 0.36/7 | 0.30/12 | 0.68/16 | 0.35/12 | 109.6M | 42.9s |
| | TimesFM (2024) | 0.30/19 | 0.74/20+ | 0.34/14 | 0.28/17 | 0.67/19 | 0.34/16 | 203.5M | 83.8s |
| | **PaAno (Ours)** | **0.53/1** | **0.89/1** | **0.49/1** | **0.47/1** | **0.87/1** | **0.52/1** | **0.3M** | **5.4s** |

Table 3: Experimental results on **multivariate time-series anomaly detection** in TSB-AD-M. For each measure, the best and second-best values are indicated in **bold** and underlined. All scores are reported with their rankings as "Score/Rank".

| | Method | VUS-PR | VUS-ROC | Range-F1 | AUC-PR | AUC-ROC | Point-F1 | # Params | Run Time |
|---|---|---|---|---|---|---|---|---|---|
| **Stat&ML** | KMeansAD (2001) | 0.29/14 | 0.73/6 | 0.33/7 | 0.25/15 | 0.69/6 | 0.31/13 | – | 62.0s |
| | PCA (2017) | 0.31/3 | 0.74/4 | 0.29/11 | 0.31/4 | 0.70/4 | 0.37/3 | – | 0.1s |
| | DLinear (2023) | 0.29/14 | 0.70/13 | 0.21/19 | 0.27/10 | 0.66/12 | 0.32/10 | < 0.1M | 14.8s |
| | NLinear (2023) | 0.30/8 | 0.70/13 | 0.21/19 | 0.26/13 | 0.65/14 | 0.31/13 | < 0.1M | 15.0s |
| **NN** | DeepAnT (2018) | 0.31/3 | 0.76/2 | 0.37/4 | 0.32/3 | 0.73/2 | 0.37/3 | < 0.1M | 9.5s |
| | OmniAnomaly (2019) | 0.31/3 | 0.69/16 | 0.37/4 | 0.27/10 | 0.65/14 | 0.32/10 | < 0.1M | 9.1s |
| | TimesNet (2022) | 0.19/20+ | 0.64/20+ | 0.17/20+ | 0.13/20+ | 0.56/20+ | 0.20/20+ | < 0.1M | 52.1s |
| | FITS (2023) | 0.21/20+ | 0.66/20+ | 0.16/20+ | 0.15/20+ | 0.58/20+ | 0.22/20+ | < 0.1M | 16.7s |
| | DADA (2025) | 0.31/3 | 0.73/6 | 0.25/14 | 0.31/4 | 0.69/6 | 0.35/6 | 1.84M | 2.1s |
| | KAN-AD (2025) | 0.40/2 | 0.76/2 | **0.41/1** | 0.35/2 | 0.73/2 | 0.40/2 | < 0.1M | 31.9s |
| **Transformer** | AnomalyTransformer (2021) | 0.12/20+ | 0.57/20+ | 0.14/20+ | 0.07/20+ | 0.52/20+ | 0.12/20+ | 4.8M | 55.8s |
| | DCdetector (2023) | 0.10/20+ | 0.54/20+ | 0.09/20+ | 0.05/20+ | 0.48/20+ | 0.10/20+ | 0.8M | 15.0s |
| | OFA (2023b) | 0.21/20+ | 0.63/20+ | 0.17/20+ | 0.15/20+ | 0.55/20+ | 0.21/20+ | 81.9M | 532.9s |
| | PatchTST (2023) | 0.30/8 | 0.71/9 | 0.22/18 | 0.28/8 | 0.67/8 | 0.33/8 | 6.9M | 66.9s |
| | iTransformer (2024) | 0.30/8 | 0.71/9 | 0.18/20+ | 0.24/16 | 0.64/19 | 0.29/16 | 6.4M | 24.4s |
| | CATCH (2025) | 0.30/8 | 0.72/8 | 0.18/20+ | 0.23/18 | 0.65/14 | 0.29/16 | 210.8M | 40.1s |
| | **PaAno (Ours)** | **0.43/1** | **0.79/1** | **0.41/1** | **0.38/1** | **0.76/1** | **0.43/1** | **0.3M** | **9.7s** |

methods, achieved the second-best results on five measures, excluding VUS-PR. Statistical and machine learning methods overall achieved competitive performance relative to those in the other categories, despite their simplicity. In particular, (Sub)-PCA recorded the second-best VUS-PR score and modest scores on three other measures. Transformer-based methods showed relatively low performance despite their heavier architectures.

In multivariate time-series anomaly detection (Table 3), PaAno again ranked first across all six performance measures. Among the baselines, several neural network-based methods showed superior performance to other methods. KAN-AD achieved the second-best VUS-PR score, while DADA, DeepAnT and OmniAnomaly ranked third. Among statistical and machine learning methods, PCA also ranked third in VUS-PR. Transformer-based methods again showed relatively low performance.

**Discussion** The results show that introducing an inductive bias toward locality and explicitly modeling short-range temporal dependencies was highly effective and efficient for time-series anomaly detection. Rather than focusing on the processing time-series in a sequential manner, PaAno treats them as collections of temporally structured patches, enabling it to capture even subtle deviations from normal patterns and thereby achieve superior performance in anomaly detection.

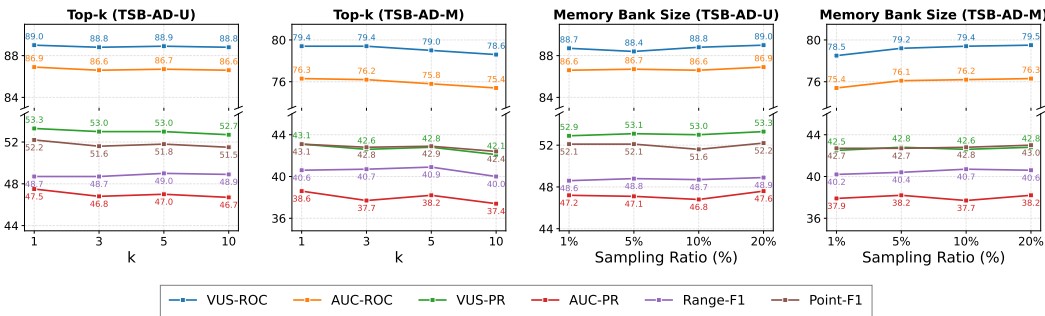

Figure 5: Sensitivity analysis on Top-$k$ and memory bank size of PaAno across TSB-AD-U/M.

In practical deployments, the patterns of normal data may change over time. PaAno can address this with a simple online update of the memory bank without requiring model retraining. By constructing the memory bank as a queue that inserts recent normal patch embeddings and discards old ones, it continually reflects up-to-date normal patterns and remains robust to non-stationary normal regimes.

**Sensitivity Analysis** We conducted a comprehensive ablation study to assess the contribution of each component in PaAno. Some important results are summarized in Table 4. Removing instance normalization and excluding either the triplet or pretext loss from the training objective lead to a substantial drop in performance, indicating that each component contributes to the effectiveness of PaAno. The triplet loss, with the

Table 4: Ablation study on the core components of PaAno. The best values for each measure are indicated in **bold**.

| | Ablation Variant | VUS-PR | VUS-ROC | Range-F1 | AUC-PR | AUC-ROC | Point-F1 |
|---|---|---|---|---|---|---|---|
| TSB-AD-U | w/o InstanceNorm | 45.3 | 85.8 | 44.6 | 41.8 | 83.8 | 46.5 |
| | w/o $\mathcal{L}_{\text{triplet}}$ and $\mathcal{L}_{\text{pretext}}$ | 48.0 | 88.1 | 45.7 | 42.5 | 85.8 | 47.7 |
| | w/o Negative Selection in $\mathcal{L}_{\text{triplet}}$ | 50.9 | 87.7 | 48.0 | 44.3 | 85.4 | 49.8 |
| | Replace $\mathcal{L}_{\text{triplet}}$ with InfoNCE loss | 48.3 | 86.1 | 45.6 | 42.3 | 83.8 | 47.4 |
| | w/o $\mathcal{L}_{\text{pretext}}$ | 51.1 | 88.9 | 47.9 | 46.0 | 86.8 | 50.7 |
| | Continuous Use of $\mathcal{L}_{\text{pretext}}$ | 47.4 | 87.3 | 46.6 | 41.1 | 84.8 | 46.6 |
| | w/o Linear Decay on $\mathcal{L}_{\text{pretext}}$ | 52.9 | 88.7 | 48.5 | 45.6 | 86.4 | 51.4 |
| | **PaAno** (Ours) | **53.0** | **88.8** | **48.7** | **46.8** | **86.6** | **51.6** |
| TSB-AD-M | w/o InstanceNorm | 33.4 | 74.7 | 39.9 | 29.4 | 73.2 | 37.5 |
| | w/o $\mathcal{L}_{\text{triplet}}$ and $\mathcal{L}_{\text{pretext}}$ | 35.6 | 76.7 | 33.1 | 30.2 | 73.5 | 36.5 |
| | w/o Negative Selection in $\mathcal{L}_{\text{triplet}}$ | 40.2 | 77.1 | 39.1 | 35.1 | 74.9 | 40.7 |
| | Replace $\mathcal{L}_{\text{triplet}}$ with InfoNCE loss | 36.2 | 76.4 | 33.5 | 31.1 | 73.1 | 35.9 |
| | w/o $\mathcal{L}_{\text{pretext}}$ | 42.2 | 79.2 | 39.6 | 37.1 | 75.9 | 42.4 |
| | Continuous Use of $\mathcal{L}_{\text{pretext}}$ | 40.8 | 77.4 | 39.8 | 36.6 | 74.3 | 41.5 |
| | w/o Linear Decay on $\mathcal{L}_{\text{pretext}}$ | 42.4 | 78.8 | 40.3 | 37.4 | 76.0 | 42.7 |
| | **PaAno** (Ours) | **42.6** | **79.4** | **40.7** | **37.7** | **76.2** | **42.8** |

farthest patch in the embedding space used as the negative pair, consistently outperforms InfoNCE and other variants. While the pretext loss in PaAno is applied only during the early stage of training, altering its scheduling to use it in the later stage degrades performance and increases computational cost. A detailed analysis of the ablation study is provided in Appendix D.

We also demonstrate that PaAno is robust to its hyperparameters, as it maintains stable performance across different settings. Figure 5 shows the results obtained by varying the memory bank size and the number of nearest neighbors used in anomaly scoring. PaAno further shows robustness to other hyperparameters, including the patch encoder architecture, loss weight, patch size and minibatch size. Full results are reported in Appendix D.

**Run Time** We measured the average run time of each method across the datasets within each benchmark. As shown in Tables 2 and 3, PaAno showed highly competitive run time, highlighting its practical efficiency for real-time applications. While majority of recent Transformer-based baselines required significantly longer run times due to their heavy architectures and resource demands, PaAno was substantially faster with superior performances. Detailed results are provided in Appendix E.

## 5 CONCLUSION

We proposed PaAno, a simple yet effective method for fast and efficient time-series anomaly detection. Instead of relying on heavy model architectures, PaAno employs a lightweight 1D-CNN to map time-series patches into vector embeddings and leverages patch-based representation learning through metric learning and a pretext task. We evaluated PaAno on the TSB-AD benchmark, which offers a rigorous evaluation protocol with performance measures that exclude point adjustment and threshold tuning. PaAno consistently achieved state-of-the-art performance compared to existing methods in both univariate and multivariate anomaly detection. Its architectural simplicity and computational efficiency make it well-suited for actual deployment in real-world industrial applications.

## ACKNOWLEDGMENTS

This work was supported by the National Research Foundation of Korea (NRF) grant funded by the Korea government (MSIT; Ministry of Science and ICT) (No. RS-2023-00207903).

## ETHICS STATEMENT

This work solely proposes a time-series anomaly detection method. It does not involve any human subjects or personally identifiable information, and we consider the risk of misuse of this work to be low.

## REPRODUCIBILITY STATEMENT

Our code is fully included in the submitted source files. To aid reproducibility, all hyperparameters and environmental details used in this paper are provided in Appendix B. All search space for other compared methods are also fully provided in Appendix B. All datasets used in this study are publicly accessible, and all information about them is contained in this paper.

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

## A PSEUDOCODE

Algorithm 1 presents the pseudocode of the training procedure. Algorithm 2 presents the pseudocode of the anomaly detection.

---

**Algorithm 1** Training Procedure of PaAno

---

**Input**: Time-series training dataset $\mathbf{X} = (\mathbf{x}_1, \ldots, \mathbf{x}_N)$
**Output**: Trained patch encoder $f_\theta$
1: $\mathcal{P} \leftarrow \{\mathbf{p}_t = (\mathbf{x}_t, \ldots, \mathbf{x}_{t+w-1})\}_{t=1}^{N-w+1}$
2: $f_\theta \leftarrow$ initialize patch encoder
3: $g_\theta \leftarrow$ initialize projection head
4: $c_\theta \leftarrow$ initialize classification head
5: **for** iteration = 1 to $T_{\text{iter}}$ **do**
6:     Sample minibatch $\mathcal{B} = \{\mathbf{p}_i\}_{i=1}^M$ from $\mathcal{P}$
7:     **for** each anchor patch $\mathbf{p}_i \in \mathcal{B}$ **do**
8:         $\mathbf{h}_i \leftarrow f_\theta(\mathbf{p}_i)$
9:         $\mathbf{z}_i \leftarrow g_\theta(\mathbf{h}_i)$
10:         $\mathbf{p}_i^+ \leftarrow$ positive patch of $\mathbf{p}_i$ from $\mathcal{P}$
11:         $\mathbf{p}_i^- \leftarrow$ farthest negative patch of $\mathbf{p}_i$ from $\mathcal{B} \setminus \{\mathbf{p}_i\}$
12:         $\mathbf{z}_i^+ \leftarrow g_\theta(f_\theta(\mathbf{p}_i^+))$
13:         $\mathbf{z}_i^- \leftarrow g_\theta(f_\theta(\mathbf{p}_i^-))$
14:         $\mathbf{p}_i^{\text{pre}} \leftarrow$ preceding patch of $\mathbf{p}_i$ from $\mathcal{P}$
15:         $\{\mathbf{p}_{i,j}^{\text{rand}}\}_{j=1}^U \leftarrow U$ random patches from $\mathcal{B} \setminus \{\mathbf{p}_i\}$
16:         $\mathbf{h}_i^{\text{pre}} \leftarrow f_\theta(\mathbf{p}_i^{\text{pre}})$
17:         $\mathbf{h}_{i,j}^{\text{rand}} \leftarrow f_\theta(\mathbf{p}_{i,j}^{\text{rand}}), \forall j \in \{1, \ldots, U\}$
18:     **end for**
19:     $\mathcal{L}_{\text{tri}} \leftarrow \frac{1}{M} \sum_{i=1}^M \max\big(0, \text{dist}(\mathbf{z}_i, \mathbf{z}_i^+) - \text{dist}(\mathbf{z}_i, \mathbf{z}_i^-) + \delta\big)$
20:     $\mathcal{L}_{\text{pre}} \leftarrow \frac{1}{M} \sum_{i=1}^M \big[-\log c_\theta(\mathbf{h}_i, \mathbf{h}_i^{\text{pre}}) - \frac{1}{U} \sum_{j=1}^U \log\big(1 - c_\theta(\mathbf{h}_i, \mathbf{h}_{i,j}^{\text{rand}})\big)\big]$
21:     Update $f_\theta$, $g_\theta$, and $c_\theta$ to minimize $\mathcal{L} = \mathcal{L}_{\text{tri}} + \lambda \cdot \mathcal{L}_{\text{pre}}$
22: **end for**
23: **return** $f_\theta$

---

**Algorithm 2** Anomaly Detection Procedure of PaAno

---

**Input**: Trained patch encoder $f_\theta$, Reduced memory bank $\hat{\mathcal{M}}$, Time-series dataset $\mathbf{X}^{\text{test}} = (\mathbf{x}_1, \ldots, \mathbf{x}_{N'})$
**Output**: Anomaly scores $\{s_{t_*}\}_{t_*=1}^{N'}$
1: **for** $t_* = 1$ to $N'$ **do**
2:     $\mathcal{P}_{t_*} \leftarrow \{\mathbf{p}_t = (\mathbf{x}_t, \ldots, \mathbf{x}_{t+w-1}) \mid t = t_* - w + 1, \ldots, t_*\}$
3:     $\mathcal{S}_{t_*} \leftarrow \emptyset$
4:     **for** each $\mathbf{p}_t \in \mathcal{P}_{t_*}$ **do**
5:         $\mathbf{h}_t \leftarrow f_\theta(\mathbf{p}_t)$
6:         $\{\mathbf{m}_t^{(1)}, \ldots, \mathbf{m}_t^{(k)}\} \leftarrow$ select the $k$ nearest neighbors of $\mathbf{h}_t$ in cosine distance from $\hat{\mathcal{M}}$
7:         $S(\mathbf{p}_t) \leftarrow \frac{1}{k} \sum_{i=1}^k \text{dist}(\mathbf{h}_t, \mathbf{m}_t^{(i)})$
8:         $\mathcal{S}_{t_*} \leftarrow \mathcal{S}_{t_*} \cup \{S(\mathbf{p}_t)\}$
9:     **end for**
10:     $s_{t_*} \leftarrow \frac{1}{|\mathcal{P}_{t_*}|} \sum_{S(\mathbf{p}_t) \in \mathcal{S}_{t_*}} S(\mathbf{p}_t)$
11: **end for**
12: **return** $\{s_{t_*}\}_{t_*=1}^{N'}$

---

## B IMPLEMENTATION DETAILS

### B.1 IMPLEMENTATION DETAILS OF PAANO

The patch encoder $f_\theta$ was a 1D-CNN consisting of four 1D convolutional layers with kernel sizes $[7, 5, 3, 3]$ and channel dimensions $[128, 256, 128, 64]$, each followed by batch normalization and a ReLU activation. A global average pooling layer was applied after the final convolutional layer to obtain a 64-dimensional patch embedding. The projection head $g_\theta$ was a two-layer MLP with ReLU

activation in the first layer. Both layers had a dimensionality of 256. The classification head $c_\theta$ was a one-layer MLP with sigmoid activation.

We adopted instance normalization (Kim et al., 2022b) following a widely used convention in recent time-series anomaly detection (Yang et al., 2023; Wu et al., 2025) and forecasting methods (Jin et al., 2024; Wang et al., 2024). For the hyperparameters, the maximum offset $r$ for defining positive patches was set to 2, and the margin $\delta$ for the triplet loss was set to 0.1. The triplet loss was divided by 10 during training. The number of per-anchor random patches $U$ was set to 5. The model was trained for 100 iterations using the AdamW optimizer with a minibatch size $M$ of 512 and a weight decay of $1e{-}4$. The learning rate was decayed to one-tenth of its initial value using a cosine annealing scheduler. The pretext loss weight $\lambda$ was linear decayed from 1 to 0 during the first 20 iterations and fixed at 0 thereafter. The memory bank size was set to $10\%$ of the original patch set $\mathcal{P}$. The number of nearest neighbors $k$ in the anomaly scoring function was set to 3. The patch size $w$ and initial learning rate were explored from $\{32, 64, 96\}$ and $\{1e{-}3, 1e{-}4, 1e{-}5\}$, respectively, based on VUS-PR performance on the Tuning split of the TSB-AD benchmark. A patch size of 96 and a learning rate of $1e{-}4$ were selected for both TSB-AD-U and TSB-AD-M.

Experiments were conducted using an NVIDIA RTX 2080Ti GPU with 11GB of memory. Each experiment was repeated 10 times with different random seeds, and the average results are reported.

### B.2 HYPERPARAMETER TUNING FOR BASELINE METHODS

Among the 46 baseline methods, 37 are applicable to univariate time-series anomaly detection and 29 to multivariate detection. The hyperparameters of the baseline methods were tuned in the same manner as PaAno, using VUS-PR performance on the Tuning split of the TSB-AD benchmark (Liu & Paparrizos, 2024). For the baseline methods included in TSB-AD, we adopted the best hyperparameter settings reported for their search spaces in the benchmark. For the remaining baseline methods, we conducted hyperparameter searches by defining comparable search spaces. The complete search spaces for all baseline methods are summarized in Tables 5 and 6.

## C   EVALUATION OF TIME-SERIES ANOMALY DETECTION

### C.1   CHALLENGES IN EVALUATION PRACTICES

The recent studies on time-series anomaly detection have often relied on evaluation protocols that introduce several biases, undermining the validity of reported results (Liu & Paparrizos, 2024; Sarfraz et al., 2024).

First, several commonly used benchmark datasets exhibit known structural flaws (Liu & Paparrizos, 2024). A primary issue is mislabeling, where inconsistencies in labeling lead to some anomaly-labeled observations being indistinguishable from normal patterns. Another common issue is unrealistic assumptions about anomaly distributions, such as assuming that anomalies occur only once or appear only at the end of a time series. These flaws compromise the reliability and validity of evaluations.

Second, the reliance on performance measures that use point adjustment and threshold tuning has created an illusion of effectiveness for sophisticated methods. Point adjustment (Kim et al., 2022a) treats an entire anomaly segment as correctly detected if even a single point within the segment is detected. While originally intended to address temporal misalignments and noisy labels, this can misleadingly inflate performance measures (Sarfraz et al., 2024; Bhattacharya et al., 2024). Threshold tuning is typically performed post-hoc and tailored to each method (Paparrizos et al., 2022; Liu & Paparrizos, 2024; Sarfraz et al., 2024). Customizing the threshold selection strategy for each method can probably lead to biased evaluations tailored to specific approaches. Also, determining a universal threshold is challenging due to varying periodicities and variances in time-series data.

### C.2   TOWARD MORE RELIABLE EVALUATION

We adopt the TSB-AD benchmark (Liu & Paparrizos, 2024), grounded in recent rigorous studies on time-series anomaly detection. This benchmark mitigates dataset-related flaws by correcting la-

Table 5: Hyperparameter search spaces for 38 univariate time-series anomaly detection methods.

| Category | Method | Hyperparameter 1 | Hyperparameter 2 |
|---|---|---|---|
| Statistical & Machine Learning | DLinear | win_size: [60, 80, 100] | None |
| | IForest | n_estimators: [25, 50, 100, 150, 200] | None |
| | KMeansAD | n_clusters: [10, 20, 30, 40] | win_size: [10, 20, 30, 40] |
| | KShapeAD | periodicity: [1, 2, 3] | None |
| | LOF | n_neighbors: [10, 20, 30, 40, 50] | metric: [minkowski, manhattan, euclidean] |
| | MatrixProfile | periodicity: [1, 2, 3] | None |
| | NLinear | win_size: [60, 80, 100] | None |
| | POLY | periodicity: [1, 2, 3] | power: [1, 2, 3, 4] |
| | SAND | periodicity: [1, 2, 3] | None |
| | Series2Graph | periodicity: [1, 2, 3] | None |
| | SR | periodicity: [1, 2, 3] | None |
| | (Sub)-HBOS | periodicity: [1, 2, 3] | n_bins: [5, 10, 20, 30, 40] |
| | (Sub)-IForest | periodicity: [1, 2, 3] | n_estimators: [25, 50, 100, 150, 200] |
| | (Sub)-KNN | periodicity: [1, 2, 3] | n_neighbors: [10, 20, 30, 40, 50] |
| | (Sub)-LOF | periodicity: [1, 2, 3] | n_neighbors: [10, 20, 30, 40, 50] |
| | (Sub)-MCD | periodicity: [1, 2, 3] | support_fraction: [0.2, 0.4, 0.6, 0.8, None] |
| | (Sub)-OCSVM | periodicity: [1, 2, 3] | kernel: [linear, poly, rbf, sigmoid] |
| | (Sub)-PCA | periodicity: [1, 2, 3] | n_components: [0.25, 0.5, 0.75, None] |
| Conventional Neural Network | AutoEncoder | win_size: [50, 100, 150] | hidden_neurons: [[64, 32], [32, 16], [128, 64]] |
| | DADA | batch_size: [32, 64, 96] | None |
| | DeepAnT | win_size: [50, 100, 150] | num_channel: [[32, 32, 40], [16, 32, 64]] |
| | Donut | win_size: [60, 90, 120] | lr: [0.001, 0.0001, 1e-05] |
| | FITS | win_size: [100, 200] | lr: [0.001, 0.0001, 1e-05] |
| | KAN-AD | win_size: [32, 64, 96] | lr: [0.01, 0.001, 0.0001] |
| | LSTMAD | win_size: [50, 100, 150] | lr: [0.0004, 0.0008] |
| | OmniAnomaly | win_size: [5, 50, 100] | lr: [0.002, 0.0002] |
| | TimesNet | win_size: [32, 96, 192] | lr: [0.001, 0.0001, 1e-05] |
| | TranAD | win_size: [5, 10, 50] | lr: [0.001, 0.0001] |
| | USAD | win_size: [5, 50, 100] | lr: [0.001, 0.0001, 1e-05] |
| Transformer | AnomalyTransformer | win_size: [50, 100, 150] | lr: [0.001, 0.0001, 1e-05] |
| | Chronos | win_size: [50, 100, 150] | None |
| | DCdetector | win_size: [80, 100] | lr: [0.0001, 1e-05] |
| | iTransformer | win_size: [64, 96] | lr: [0.0001, 5e-05] |
| | Lag-Llama | win_size: [32, 64, 96] | None |
| | MOMENT (FT) | win_size: [64, 128, 256] | None |
| | MOMENT (ZS) | win_size: [64, 128, 256] | None |
| | OFA | win_size: [50, 100, 150] | None |
| | PatchTST | num_epoch: [5, 10, 15] | None |
| | TimesFM | win_size: [32, 64, 96] | None |
| Ours | PaAno | patch_size: [32, 64, 96] | lr: [0.001, 0.0001, 1e-05] |

beling inconsistencies and reflecting realistic anomaly distributions. It also systematically addresses issues such as point adjustment and threshold tuning, enabling fair and consistent evaluation.

Regarding performance measures for time-series anomaly detection, Paparrizos et al. (2022) proposed the VUS of the Precision-Recall Curve (VUS-PR) and the Receiver Operating Characteristic curve (VUS-ROC) as robust and lag-tolerant measures, with resilience to temporal misalignment and noise. Building upon this, Liu & Paparrizos (2024) empirically validated VUS-PR as the most fair and reliable measure, capable of jointly capturing detection accuracy and localization quality at the segment level. Sarfraz et al. (2024) recommended the joint use of Point-F1 and Range-F1, both of which are threshold-dependent measures, and suggested supplementing them with the Area Under the PR Curve (AUC-PR) as a threshold-independent measure. Additionally, the Area Under the ROC Curve (AUC-ROC) remains one of the most widely used measures for providing a global view of ranking performance, despite being sensitive to random scores in highly imbalanced settings (Paparrizos et al., 2022; Liu & Paparrizos, 2024). Overall, employing multiple complementary measures is essential for a comprehensive and reliable evaluation.

Following this guidance, we adopted six performance measures in this study: VUS-PR, VUS-ROC, and Range-F1 as range-wise measures; and AUC-PR, AUC-ROC, and Point-F1 as point-wise measures. Four measures—excluding Range-F1 and Point-F1—serve as threshold-independent measures, and point adjustment was not applied to any of the measures.

Table 6: Hyperparameter search spaces for 30 multivariate time-series anomaly detection methods.

| Category | Method | Hyperparameter 1 | Hyperparameter 2 |
|---|---|---|---|
| | CBLOF | n_clusters: [4, 8, 16, 32] | alpha: [0.6, 0.7, 0.8, 0.9] |
| | COPOD | None | None |
| | DLinear | win_size: [60, 80, 100] | None |
| | EIF | n_trees: [25, 50, 100, 200] | None |
| | HBOS | n_bins: [5, 10, 20, 30, 40] | tol: [0.1, 0.3, 0.5, 0.7] |
| | IForest | n_estimators: [25, 50, 100, 150, 200] | max_features: [0.2, 0.4, 0.6, 0.8, 1.0] |
| Statistical & | KMeansAD | n_clusters: [10, 20, 30, 40] | window_size: [10, 20, 30, 40] |
| Machine Learning | KNN | n_neighbors: [10, 20, 30, 40, 50] | method: [largest, mean, median] |
| | LOF | n_neighbors: [10, 20, 30, 40, 50] | metric: [minkowski, manhattan, euclidean] |
| | MCD | support_fraction: [0.2, 0.4, 0.6, 0.8, None] | None |
| | NLinear | win_size: [60, 80, 100] | None |
| | OCSVM | kernel: [linear, poly, rbf, sigmoid] | nu: [0.1, 0.3, 0.5, 0.7] |
| | PCA | n_components: [0.25, 0.5, 0.75, None] | None |
| | RobustPCA | max_iter: [500, 1000, 1500] | None |
| | AutoEncoder | win_size: [50, 100, 150] | hidden_neurons: [[64, 32], [32, 16], [128, 64]] |
| | DADA | batch_size: [32, 64, 96] | None |
| | DeepAnT | win_size: [50, 100, 150] | num_channel: [[32, 32, 40], [16, 32, 64]] |
| | Donut | win_size: [60, 90, 120] | lr: [0.001, 0.0001, 1e-05] |
| | FITS | win_size: [100, 200] | lr: [0.001, 0.0001, 1e-05] |
| Conventional | KAN-AD | win_size: [32, 64, 96] | lr: [0.01, 0.001, 0.0001] |
| Neural Network | LSTMAD | win_size: [50, 100, 150] | lr: [0.0004, 0.0008] |
| | OmniAnomaly | win_size: [5, 50, 100] | lr: [0.002, 0.0002] |
| | TimesNet | win_size: [32, 96, 192] | lr: [0.001, 0.0001, 1e-05] |
| | TranAD | win_size: [5, 10, 50] | lr: [0.001, 0.0001] |
| | USAD | win_size: [5, 50, 100] | lr: [0.001, 0.0001, 1e-05] |
| | AnomalyTransformer | win_size: [50, 100, 150] | lr: [0.001, 0.0001, 1e-05] |
| | CATCH | patch_size: [16, 32, 64] | lr: [0.0001, 5e-05] |
| | DCdetector | win_size: [80, 100] | lr: [0.0001, 1e-05] |
| Transformer | iTransformer | win_size: [64, 96] | lr: [0.0001, 5e-05] |
| | OFA | win_size: [50, 100, 150] | None |
| | PatchTST | num_epoch: [5, 10, 15] | None |
| Ours | PaAno | patch_size: [32, 64, 96] | lr: [0.001, 0.0001, 1e-05] |

## C.3 DETAILS OF PERFORMANCE MEASURES

**Notations** Let $\mathbf{X}' = (\mathbf{x}_1, \ldots, \mathbf{x}_{N'})$ denote the test dataset consisting of $N'$ time steps, where each $\mathbf{x}_t \in \mathbb{R}^d$ represents a $d$-dimensional observation at time $t$. Let $y_t \in \{0, 1\}$ denote the corresponding ground-truth label, where $y_t = 1$ indicates an anomaly and $y_t = 0$ otherwise. Let $s_t \in \mathbb{R}$ denote the predicted anomaly score at time $t$, and $\hat{y}_t(\tau) = \mathbf{1}(s_t \geq \tau)$ the binarized prediction obtained by thresholding the score at threshold $\tau$.

**AUC-ROC** The Receiver Operating Characteristic (ROC) curve plots the true positive rate (TPR) against the false positive rate (FPR) as the threshold $\tau$ varies. The TPR and FPR at threshold $\tau$ are defined as:

$$\text{TPR}(\tau) = \frac{\sum_{t=1}^{N'} \hat{y}_t(\tau) \cdot y_t}{\sum_{t=1}^{N'} y_t}; \quad \text{FPR}(\tau) = \frac{\sum_{t=1}^{N'} \hat{y}_t(\tau) \cdot (1 - y_t)}{\sum_{t=1}^{N'} (1 - y_t)}.$$

Given a finite set of thresholds $\{\tau_1, \tau_2, \ldots, \tau_K\}$ sorted in descending order, the Area Under the ROC Curve (AUC-ROC) is computed as:

$$\text{AUC-ROC} = \sum_{k=1}^{K-1} \left( \text{FPR}(\tau_k) - \text{FPR}(\tau_{k+1}) \right) \cdot \frac{\text{TPR}(\tau_k) + \text{TPR}(\tau_{k+1})}{2}$$

**AUC-PR** The Area Under the Precision-Recall Curve (AUC-PR) summarizes the trade-off between precision and recall across varying thresholds:

$$\text{AUC-PR} = \sum_{k=1}^{K-1} \left( R(\tau_k) - R(\tau_{k+1}) \right) \cdot \frac{P(\tau_k) + P(\tau_{k+1})}{2},$$

where $P(\tau)$ and $R(\tau)$ denote the precision and recall at threshold $\tau$, computed as:

$$\text{P}(\tau) = \frac{\sum_{t=1}^{N'} \hat{y}_t(\tau) \cdot y_t}{\sum_{t=1}^{N'} \mathbf{1}(s_t \geq \tau)}; \quad \text{R}(\tau) = \frac{\sum_{t=1}^{N'} \hat{y}_t(\tau) \cdot y_t}{\sum_{t=1}^{N'} y_t}.$$

**Point-F1** The standard point-wise F1 score (Point-F1) is defined as the maximum F1 score over all possible thresholds $\tau$:

$$\text{Point-F1} = \max_\tau \frac{2 \cdot \text{P}(\tau) \cdot \text{R}(\tau)}{\text{P}(\tau) + \text{R}(\tau)}.$$

**Range-F1** An *anomaly segment* is defined as a contiguous subsequence of time steps labeled as anomaly. Let $\mathcal{A} = \{A_1, \ldots, A_M\}$ denote the set of ground-truth segments, where $A_j = \{t \mid a_j \leq t \leq b_j, \ y_t = 1\}$. For a given threshold $\tau$, let $\hat{\mathcal{A}}(\tau) = \{\hat{A}_1, \ldots, \hat{A}_{N_\tau}\}$ denote the set of predicted segments, where $\hat{A}_i = \{t \mid \hat{a}_i \leq t \leq \hat{b}_i, \ \hat{y}_t(\tau) = 1\}$.

$$\text{Range-P}(\tau) = \frac{|\{\hat{A}_i \in \hat{\mathcal{A}}(\tau) \mid \exists A_j \in \mathcal{A}, \ \hat{A}_i \cap A_j \neq \emptyset\}|}{|\hat{\mathcal{A}}(\tau)|};$$

$$\text{Range-R}(\tau) = \frac{|\{A_j \in \mathcal{A} \mid \exists \hat{A}_i \in \hat{\mathcal{A}}(\tau), \ A_j \cap \hat{A}_i \neq \emptyset\}|}{|\mathcal{A}|}.$$

Then the range-based F1 score is:

$$\text{Range-F1} = \max_\tau \frac{2 \cdot \text{Range-P}(\tau) \cdot \text{Range-R}(\tau)}{\text{Range-P}(\tau) + \text{Range-R}(\tau)}.$$

**VUS-ROC** The Volume Under the ROC Surface (VUS-ROC) extends the standard ROC-AUC to a three-dimensional evaluation by jointly varying the threshold $\tau$ the lag tolerance $\ell$ around labeled anomalies. ROC curves are computed for all possible pairs $(\tau, \ell)$, where $\tau \in \{\tau_1, \ldots, \tau_K\}$ and $\ell \in \{0, \ldots, L\}$, forming a surface. VUS-ROC is then defined as the volume under this ROC surface.

Following the TSB-AD benchmark (Liu & Paparrizos, 2024), we use the optimized implementation of VUS proposed by Paparrizos et al. (2022). The maximum lag tolerance $L$ is determined by identifying the first prominent local maximum in the autocorrelation of the first channel of the time series, which typically corresponds to the dominant repeating interval.

**VUS-PR** Analogous to VUS-ROC, the Volume Under the Precision–Recall Surface (VUS-PR) constructs a surface by varying both the threshold $\tau$ and the lag tolerance $\ell$ over PR curves. The VUS-PR is defined as the volume under this surface, providing a threshold-independent and lag-tolerant generalization of the standard PR-AUC.

## D  SENSITIVITY ANALYSIS

We conducted a comprehensive sensitivity analysis to validate the effectiveness and robustness of the core components of PaAno.

### D.1  PATCH ENCODER ARCHITECTURE

The patch encoder $f_\theta$ in PaAno primarily adopts a simple 1D-CNN as the default in this study, but it can be flexibly extended to various other architectures. We conducted experiments comparing it with more complex encoders, including 1D-ResNet (Wang et al., 2017), Temporal Convolutional Network (TCN) (Bai et al., 2018), and OmniScaleCNN (OmniCNN) (Tang et al., 2022), which were implemented using the *tsai* package (Oguiza, 2023). To analyze the architectural sensitivity of the patch encoder, we also evaluated variants of the default 1D-CNN (4-layer, 1x width) by modifying its width (0.5× and 2×) and depth (3-layer and 5-layer) while keeping the kernel configurations unchanged.

Table 7 summarizes the results and the number of parameters of each model, with the simple 1D-CNN highlighted in bold as the default setting. In TSB-AD-U, the lighter variants of the 1D-CNN, those with reduced width (0.5x width) or depth (3-layer), achieved performance comparable to or slightly exceeding that of the heavier variants, despite having fewer parameters. Consequently, for univariate anomaly detection, a lighter 1D-CNN offers an efficient and effective choice without

Table 7: Sensitivity analysis on the patch encoder architecture. The best and second-best values for each measure are indicated in **bold** and underlined, respectively. All values are reported in percentage (%).

| | Encoder | #Params | VUS-PR | VUS-ROC | Range-F1 | AUC-PR | AUC-ROC | Point-F1 |
|---|---|---|---|---|---|---|---|---|
| | 1D-CNN (0.5× width) | 147K | 52.4 | **88.9** | 47.6 | 46.0 | **86.7** | 50.9 |
| | 1D-CNN (3 layer) | 297K | 52.6 | 88.7 | **48.8** | 46.7 | 86.5 | **51.6** |
| | **1D-CNN** | 371K | **53.0** | 88.8 | 48.7 | 46.8 | 86.6 | **51.6** |
| TSB-AD-U | 1D-CNN (5 layer) | 1126K | 52.7 | 88.2 | 48.0 | 46.4 | 86.1 | 51.3 |
| | 1D-CNN (2× width) | 1250K | 52.0 | 88.1 | 47.7 | 45.8 | 86.0 | 50.6 |
| | OmniCNN | 401K | 46.4 | 85.2 | 45.6 | 41.5 | 83.3 | 46.3 |
| | TCN | 655K | 47.2 | 86.1 | 45.5 | 42.0 | 84.1 | 47.2 |
| | 1D-ResNet | 1456K | 51.5 | 86.4 | 44.6 | 46.1 | 85.7 | 51.0 |
| | 1D-CNN (0.5× width) | 162K | 43.1 | 79.7 | 40.7 | **39.2** | 76.5 | **43.3** |
| | 1D-CNN (3 layer) | 318K | 42.1 | 79.5 | 40.7 | 37.7 | 76.3 | 42.7 |
| | **1D-CNN** | 386K | 42.6 | 79.4 | 40.7 | 37.7 | 76.2 | 42.8 |
| TSB-AD-M | 1D-CNN (5 layer) | 1148K | 42.4 | 78.6 | 39.9 | 37.3 | 75.3 | 42.7 |
| | 1D-CNN (2× width) | 1326K | **43.6** | 79.4 | 41.1 | 38.7 | 76.1 | 43.1 |
| | OmniCNN | 968K | 39.0 | 79.0 | 44.1 | 36.0 | 78.0 | 42.7 |
| | TCN | 664K | 37.9 | **80.9** | **44.7** | 35.0 | **79.0** | 42.6 |
| | 1D-ResNet | 1485K | 41.5 | 79.2 | 40.1 | 36.7 | 75.9 | 42.1 |

compromising accuracy. In contrast, in TSB-AD-M, PaAno maintained robust performance across different 1D-CNN configurations, including both lighter and heavier variants. This indicates that PaAno remains effective under width and depth variations within the 1D-CNN architecture.

For other architectures, OmniCNN showed lower performance than the default 1D-CNN in TSB-AD-U, but achieved moderate performance on Range-F1 and AUC-ROC in TSB-AD-M. TCN also showed lower performance in TSB-AD-U, but achieved higher performance on VUS-ROC, AUC-ROC, and Range-F1 in TSB-AD-M. 1D-ResNet showed lower performance than the default 1D-CNN in TSB-AD-U and provided no clear advantage in TSB-AD-M, despite requiring nearly four times more parameters. These results support the use of the 1D-CNN as the default patch encoder, as it achieves strong VUS-PR performance in both univariate and multivariate anomaly detection, while requiring fewer parameters than the alternative architectures.

## D.2 LOSS FUNCTION

Table 8: Sensitivity analysis of the triplet loss compared with the InfoNCE loss and different negative selection strategies. The best and second-best values for each measure are shown in **bold** and underline, respectively. All values are reported in percentage (%).

| | Loss function | Negative Sampling | VUS-PR | VUS-ROC | Range-F1 | AUC-PR | AUC-ROC | Point-F1 |
|---|---|---|---|---|---|---|---|---|
| | InfoNCE | All Non-Self Patch | 48.3 | 86.1 | 45.6 | 42.3 | 83.8 | 47.4 |
| | Triplet | Random | 50.9 | 87.7 | 48.0 | 44.3 | 85.4 | 49.8 |
| TSB-AD-U | Triplet | Closest | 51.6 | 88.7 | 48.6 | 45.3 | 86.5 | 50.3 |
| | Triplet | Median | 52.8 | 88.6 | 48.5 | 46.2 | 86.2 | 50.8 |
| | **Triplet** | **Farthest** | **53.0** | **88.8** | **48.7** | **46.8** | **86.6** | **51.6** |
| | InfoNCE | All Non-Self Patch | 36.2 | 76.4 | 33.5 | 31.1 | 73.1 | 35.9 |
| | Triplet | Random | 40.2 | 77.1 | 39.1 | 35.1 | 74.9 | 40.7 |
| TSB-AD-M | Triplet | Closest | 41.2 | 78.3 | 39.8 | 36.9 | 75.2 | 41.3 |
| | Triplet | Median | 41.6 | 78.8 | 39.6 | 37.2 | 75.8 | 42.2 |
| | **Triplet** | **Farthest** | **42.6** | **79.4** | **40.7** | **37.7** | **76.2** | **42.8** |

Table 8 reports the sensitivity analysis of the triplet loss in PaAno compared to InfoNCE and other negative sampling strategies. Since PaAno extracts patches through a sliding window, there is no guarantee that other patches in the minibatch form semantically meaningful negatives for a given anchor. This lack of semantic correspondence makes many InfoNCE negatives ambiguous, which weakens the contrastive signal and destabilizes the embedding space (Wang & Dou, 2023).

We demonstrated other ways to choose the negative patch to analyze how the negative selection and the triplet loss in PaAno respond to different forms of contrast. Specifically, we compared the patch farthest from the anchor in the embedding space (*Farthest*), the patch closest to the anchor (*Closest*), a randomly chosen patch (*Random*), and the patch at the median of the similarity ranking (*Median*).

In the experiments, random negatives slightly outperformed InfoNCE but exhibited similar behavior, providing limited contrast. These results suggest that random negatives fall short of providing

the meaningful contrast required for effective metric learning of time-series patches. In contrast, as the default strategy in PaAno, the farthest negative consistently achieved the strongest performance. It provides a reliably distinct comparison that encourages the encoder to learn discriminative representations. The closest and median negatives also achieved moderate performance in both univariate and multivariate settings, suggesting that explicitly selected negatives can still provide useful learning signals, although less consistently than the farthest negative. Overall, these results indicate that metric learning for time-series patches benefits from stable and informative negative selection, while the farthest negative offers the most reliable contrast for learning discriminative patch representations.

Table 9: Sensitivity analysis of the loss weight $\lambda$ with respect to the applied ratio and the scheduling strategy. Here, each ratio (e.g., 20%, 50%, 100%) denotes the proportion of the initial part of training during which the pretext loss is applied. For a specified initial ratio of the training iterations, *Linear* decays $\lambda$ linearly to zero, whereas *Constant* keeps it fixed. The best and second-best values for each measure are shown in **bold** and underline, respectively. All values are reported in percentage (%).

| | Ratio | Schedule | VUS-PR | VUS-ROC | Range-F1 | AUC-PR | AUC-ROC | Point-F1 |
|---|---|---|---|---|---|---|---|---|
| **TSB-AD-U** | **20%** | **Linear** | **53.0** | **88.8** | **48.7** | **46.8** | **86.6** | **51.6** |
| | 20% | Constant | 52.9 | 88.7 | 48.5 | 45.6 | 86.4 | 51.4 |
| | 50% | Linear | 49.4 | 87.8 | 47.7 | 43.2 | 85.6 | 48.4 |
| | 50% | Constant | 48.9 | 87.9 | 47.2 | 42.7 | 85.5 | 48.1 |
| | 100% | Linear | 47.4 | 87.3 | 46.6 | 41.1 | 84.8 | 46.6 |
| | 100% | Constant | 46.8 | 87.1 | 45.8 | 40.5 | 84.5 | 45.8 |
| **TSB-AD-M** | **20%** | **Linear** | **42.6** | **79.4** | **40.7** | **37.7** | **76.2** | **42.8** |
| | 20% | Constant | 42.4 | 78.8 | 40.3 | 37.4 | 76.0 | 42.7 |
| | 50% | Linear | 41.6 | 78.2 | 39.9 | 37.3 | 75.1 | 42.2 |
| | 50% | Constant | 41.2 | 77.7 | 40.1 | 36.8 | 74.6 | 41.6 |
| | 100% | Linear | 40.8 | 77.4 | 39.8 | 36.6 | 74.3 | 41.5 |
| | 100% | Constant | 40.4 | 77.4 | 39.4 | 36.1 | 74.3 | 41.4 |

Table 10: Sensitivity analysis of the loss weight $\lambda$. The best and second-best values for each measure are indicated in **bold** and underline, respectively. All values are reported in percentage (%).

| | $\lambda$ | VUS-PR | VUS-ROC | Range-F1 | AUC-PR | AUC-ROC | Point-F1 |
|---|---|---|---|---|---|---|---|
| **TSB-AD-U** | 0.1 | 53.1 | 88.9 | 48.8 | 47.2 | **86.8** | 52.0 |
| | 0.5 | **53.2** | **89.0** | 48.9 | **47.3** | 86.5 | **52.2** |
| | **1.0** | 53.0 | 88.8 | 48.7 | 46.8 | 86.6 | 51.6 |
| | 2.0 | 53.0 | 88.9 | **49.0** | 47.2 | 86.5 | 51.8 |
| **TSB-AD-M** | 0.1 | **42.8** | 79.0 | **40.7** | 38.1 | 75.9 | 42.8 |
| | 0.5 | 42.6 | 79.0 | 40.4 | **38.2** | 75.8 | 42.9 |
| | **1.0** | 42.6 | **79.4** | **40.7** | 37.7 | **76.2** | 42.8 |
| | 2.0 | 42.5 | 79.1 | 40.4 | 38.0 | 76.0 | **43.0** |

The pretext loss in PaAno is applied only during the early iterations to stabilize representation learning when the embedding space is still unstructured and triplet-based distance comparisons are unreliable. Using this auxiliary objective in the initial phase improves training stability and shows consistent performance gains (Table 4). Additionally, to analyze how the pretext loss should be integrated with the triplet objective, we conducted sensitivity analyses on its application strategy and loss weight.

Table 9 reports the effect of the ratio of initial iteration the pretext loss applied with scheduling methods. Applying the pretext loss only in the initial 20% portion of training yielded the best overall performance, while extending it beyond that ratio led to progressively worse results. The pretext loss is shown to interfere with triplet-based discrimination once the embedding space becomes more structured. Also, a linear-decay schedule, which gradually reduces the pretext loss instead of removing it abruptly, consistently outperformed a constant schedule across all ratios. These results shows the effectiveness of the default application strategy for pretext loss in PaAno.

Table 10 further analyzes the loss weight. The performance remains stable across a range of weights, indicating that PaAno is robust to the choice of the loss weight.

## D.3 MEMORY BANK SIZE

Table 11: Sensitivity analysis on the size of the memory bank, expressed as a percentage of the training dataset size. The best and second-best values for each measure are indicated in **bold** and underlined, respectively. All values are reported in percentage (%).

|  | Size | VUS-PR | VUS-ROC | Range-F1 | AUC-PR | AUC-ROC | Point-F1 |
|---|---|---|---|---|---|---|---|
| TSB-AD-U | 1% | 52.9 | 88.7 | 48.6 | 47.2 | 86.6 | 52.1 |
| | 5% | 53.1 | 88.4 | 48.8 | 47.1 | 86.7 | 52.1 |
| | **10%** | 53.0 | 88.8 | 48.7 | 46.8 | 86.6 | 51.6 |
| | 20% | **53.3** | **89.0** | **48.9** | **47.6** | **86.9** | **52.2** |
| TSB-AD-M | 1% | 42.5 | 78.5 | 40.2 | 37.9 | 75.4 | 42.7 |
| | 5% | **42.8** | 79.2 | 40.4 | **38.2** | 76.1 | 42.7 |
| | **10%** | 42.6 | 79.4 | **40.7** | 37.7 | 76.2 | 42.8 |
| | 20% | **42.8** | 79.5 | 40.6 | **38.2** | 76.3 | 43.0 |

Table 11 presents the results of a sensitivity analysis on the size of the memory bank $\mathcal{M}$, with the 10% highlighted in bold as the default setting. Varying the memory bank size had no significant impact on performance. Since PaAno maintains its effectiveness even with a small memory bank (*e.g.*, 1%), the size can be reduced from the default setting when fast inference or memory efficiency is required.

## D.4 NUMBER OF NEAREST NEIGHBORS USED IN ANOMALY SCORING

Table 12: Sensitivity analysis on the number of nearest neighbors retrieved from the memory bank for anomaly scoring. The best and second-best values for each measure are indicated in **bold** and underlined, respectively.

|  | $k$ | VUS-PR | VUS-ROC | Range-F1 | AUC-PR | AUC-ROC | Point-F1 |
|---|---|---|---|---|---|---|---|
| TSB-AD-U | 1 | **53.3** | **89.0** | 48.7 | **47.5** | **86.9** | **52.2** |
| | **3** | 53.0 | 88.8 | 48.7 | 46.8 | 86.6 | 51.6 |
| | 5 | 53.0 | 88.9 | **49.0** | 47.0 | 86.7 | 51.8 |
| | 10 | 52.7 | 88.8 | 48.9 | 46.7 | 86.6 | 51.5 |
| TSB-AD-M | 1 | **43.1** | **79.4** | 40.6 | **38.6** | **76.3** | **43.1** |
| | **3** | 42.6 | **79.4** | 40.7 | 37.7 | 76.2 | 42.8 |
| | 5 | 42.8 | 79.0 | **40.9** | 38.2 | 75.8 | 42.9 |
| | 10 | 42.1 | 78.6 | 40.0 | 37.4 | 75.4 | 42.4 |

Table 12 presents the results of sensitivity analysis on the number of nearest neighbors $k$ retrieved from the memory bank $\mathcal{M}$ for anomaly scoring, with the 3 highlighted in bold as the default setting. The results show that performance remained stable as $k$ varied. This suggests that PaAno is robust to the choice of $k$ around the default setting. Although PaAno shows stable performance with $k=1$, we adopt $k=3$ as the default to enhance stability against potential noise.

## D.5 PATCH SIZE

Table 13 presents the results of the sensitivity analysis on patch size. The results show that performance remains stable as the patch size varies. In both TSB-AD-U and TSB-AD-M, PaAno achieves strong performance at a patch size of 32 and shows improved performance from 64 onward. A patch size of 96 was selected for both TSB-AD-U and TSB-AD-M, based on results from the TSB-AD-Tuning set.

Table 13: Sensitivity analysis on the size of patch length. The best and second-best values for each measure are indicated in **bold** and underlined, respectively. All values are reported in percentage (%).

| | Size | VUS-PR | VUS-ROC | Range-F1 | AUC-PR | AUC-ROC | Point-F1 |
|---|---|---|---|---|---|---|---|
| **TSB-AD-U** | 32 | 44.4 | 85.7 | 42.4 | 38.6 | 81.6 | 44.3 |
| | 64 | 51.2 | 88.2 | 47.9 | 45.3 | 85.3 | 50.3 |
| | **96** | 53.0 | **88.8** | 48.7 | **46.8** | **86.6** | **51.6** |
| | 128 | **53.1** | 88.3 | **48.8** | 45.8 | 86.4 | 50.6 |
| **TSB-AD-M** | 32 | 36.6 | 75.3 | 34.3 | 31.9 | 70.2 | 36.4 |
| | 64 | 41.0 | 78.4 | 38.9 | 36.2 | 74.7 | 40.5 |
| | **96** | **42.6** | **79.4** | **40.7** | **37.7** | **76.2** | **42.8** |
| | 128 | 40.2 | 78.2 | 39.5 | 35.7 | 75.4 | 41.0 |

Table 14: Sensitivity analysis on the size of minibatch size. The best and second-best values for each measure are indicated in **bold** and underlined, respectively. All values are reported in percentage (%).

| | Size | VUS-PR | VUS-ROC | Range-F1 | AUC-PR | AUC-ROC | Point-F1 |
|---|---|---|---|---|---|---|---|
| **TSB-AD-U** | 128 | 52.8 | 88.6 | 48.6 | 46.7 | 86.5 | 51.7 |
| | 256 | 53.1 | 88.4 | **49.1** | 46.8 | **86.6** | 51.4 |
| | **512** | 53.0 | **88.8** | 48.7 | 46.8 | **86.6** | 51.6 |
| | 1024 | **53.2** | 88.7 | 48.9 | **46.9** | 86.5 | **51.8** |
| **TSB-AD-M** | 128 | 42.3 | 79.0 | 39.9 | 37.6 | 75.9 | 42.3 |
| | 256 | **42.7** | 79.2 | **41.2** | **37.8** | 76.0 | 42.6 |
| | **512** | 42.6 | **79.4** | 40.7 | 37.7 | **76.2** | **42.8** |
| | 1024 | 42.6 | 79.2 | 40.3 | 37.7 | 76.0 | 42.5 |

## D.6 Minibatch Size

Table 14 presents the results of the sensitivity analysis on minibatch size. Unlike other contrastive losses (e.g., InfoNCE), the loss function in PaAno does not heavily depend on the minibatch size, because each anchor requires only a single farthest negative pair. Across both TSB-AD-U and TSB-AD-M, PaAno remains consistently robust for all tested minibatch sizes.

## E  Detailed Experimental Results

### E.1  Subset-Wise Experimental Results of PaAno

To ensure a thorough and transparent evaluation, we present the detailed subset-wise experimental results of PaAno on TSB-AD-U and TSB-AD-M in Table 15.

### E.2  Entire Experiment Results

We report the complete experimental results for 40 methods, including PaAno, on TSB-AD-U in Table 16, and for 32 methods on TSB-AD-M in Table 17.

### E.3  Run Time

To evaluate the practical applicability of real-time anomaly detection, we measured the run time of each method, including both training and inference, averaged across the datasets within each benchmark. The results for the baseline methods are taken from the TSB-AD benchmark (Liu & Paparrizos, 2024), where statistical and machine learning methods were executed on an AMD EPYC 7713 CPU, while neural network-based and Transformer-based methods were run on an NVIDIA A100 GPU. For PaAno, we used an NVIDIA RTX 2080 Ti GPU to measure its average run time.

Table 15: Experimental results of PaAno by subset from TSB-AD-U and TSB-AD-M benchmarks.

| | Name of Subset | # Time-Series | Avg. Dim | VUS-PR | VUS-ROC | Range-F1 | AUC-PR | AUC-ROC | Point-F1 |
|---|---|---|---|---|---|---|---|---|---|
| TSB-AD-U | UCR | 70 | 1 | 0.3707±0.0101 | 0.9252±0.0030 | 0.4488±0.0070 | 0.3670±0.0120 | 0.9176±0.0034 | 0.4235±0.0071 |
| | YAHOO | 30 | 1 | 0.7011±0.0133 | 0.9481±0.0039 | 0.3201±0.0162 | 0.5847±0.0230 | 0.9458±0.0047 | 0.6263±0.0191 |
| | WSD | 20 | 1 | 0.5118±0.0121 | 0.9337±0.0042 | 0.5845±0.0107 | 0.4492±0.0115 | 0.9247±0.0044 | 0.4864±0.0134 |
| | CATSv2 | 1 | 1 | 0.3196±0.0047 | 0.7550±0.0022 | 0.1986±0.0060 | 0.4890±0.0010 | 0.7485±0.0019 | 0.5396±0.0017 |
| | Daphnet | 1 | 1 | 0.3370±0.0640 | 0.9143±0.0144 | 0.3430±0.0320 | 0.3381±0.0761 | 0.9214±0.0133 | 0.4354±0.0478 |
| | Exathlon | 30 | 1 | 0.8271±0.0053 | 0.9640±0.0016 | 0.6395±0.0120 | 0.8206±0.0050 | 0.9625±0.0016 | 0.8380±0.0054 |
| | IOPS | 15 | 1 | 0.3199±0.0220 | 0.8959±0.0070 | 0.3985±0.0099 | 0.2284±0.0191 | 0.8663±0.0119 | 0.3137±0.0218 |
| | LTDB | 8 | 1 | 0.7000±0.0058 | 0.8176±0.0047 | 0.6790±0.0115 | 0.6325±0.0064 | 0.7895±0.0048 | 0.6662±0.0061 |
| | MGAB | 8 | 1 | 0.2749±0.0109 | 0.9542±0.0056 | 0.4185±0.0133 | 0.2698±0.0101 | 0.9542±0.0055 | 0.3525±0.0063 |
| | MITDB | 7 | 1 | 0.4382±0.0132 | 0.9087±0.0056 | 0.4214±0.0156 | 0.3734±0.0161 | 0.8686±0.0062 | 0.4405±0.0157 |
| | MSL | 7 | 1 | 0.2527±0.0082 | 0.7159±0.0083 | 0.3783±0.0127 | 0.2283±0.0141 | 0.6701±0.0098 | 0.3281±0.0114 |
| | NAB | 23 | 1 | 0.5040±0.0104 | 0.7864±0.0090 | 0.5771±0.0089 | 0.4907±0.0116 | 0.7682±0.0095 | 0.5404±0.0116 |
| | NEK | 8 | 1 | 0.5614±0.0209 | 0.7394±0.0166 | 0.5944±0.0146 | 0.5906±0.0226 | 0.7638±0.0098 | 0.6347±0.0189 |
| | OPPORTUNITY | 27 | 1 | 0.2698±0.0252 | 0.6920±0.0196 | 0.3663±0.0191 | 0.2763±0.0266 | 0.6866±0.0208 | 0.3410±0.0228 |
| | Power | 1 | 1 | 0.1898±0.0087 | 0.6634±0.0067 | 0.2491±0.0163 | 0.1953±0.0182 | 0.6427±0.0067 | 0.2744±0.0064 |
| | SED | 2 | 1 | 0.9492±0.0089 | 0.9984±0.0003 | 0.8344±0.0144 | 0.7940±0.0323 | 0.9937±0.0008 | 0.8159±0.0151 |
| | SMAP | 17 | 1 | 0.8065±0.0044 | 0.9199±0.0018 | 0.7738±0.0053 | 0.7931±0.0047 | 0.9167±0.0021 | 0.7709±0.0033 |
| | SMD | 33 | 1 | 0.4935±0.0285 | 0.9218±0.0055 | 0.5490±0.0230 | 0.4603±0.0321 | 0.9207±0.0062 | 0.5253±0.0233 |
| | Stock | 8 | 1 | 0.7164±0.0032 | 0.8414±0.0009 | 0.1257±0.0327 | 0.0821±0.0009 | 0.4927±0.0027 | 0.1511±0.0003 |
| | SVDB | 18 | 1 | 0.7069±0.0103 | 0.9813±0.0010 | 0.6203±0.0104 | 0.6417±0.0106 | 0.9704±0.0012 | 0.6613±0.0059 |
| | SWaT | 1 | 1 | 0.0962±0.0008 | 0.1758±0.0052 | 0.1136±0.0071 | 0.0965±0.0010 | 0.1954±0.0045 | 0.2153±0.0000 |
| | TAO | 2 | 1 | 0.8748±0.0049 | 0.9341±0.0011 | 0.2025±0.1105 | 0.1156±0.0049 | 0.5035±0.0116 | 0.2099±0.0005 |
| | TODS | 13 | 1 | 0.7301±0.0030 | 0.8760±0.0011 | 0.2946±0.0180 | 0.3752±0.0061 | 0.7856±0.0010 | 0.4181±0.0076 |
| **TSB-AD-U Average** | | | | **0.5296±0.0027** | **0.8877±0.0012** | **0.4869±0.0030** | **0.4682±0.0038** | **0.8660±0.0011** | **0.5164±0.0032** |
| TSB-AD-M | CATSv2 | 5 | 17 | 0.0690±0.0072 | 0.6881±0.0088 | 0.1381±0.0127 | 0.0744±0.0117 | 0.6813±0.0105 | 0.1285±0.0145 |
| | CreditCard | 1 | 29 | 0.0343±0.0182 | 0.5558±0.0757 | 0.0351±0.0179 | 0.0068±0.0039 | 0.5652±0.0875 | 0.0416±0.0151 |
| | Daphnet | 1 | 9 | 0.2706±0.0489 | 0.8925±0.0192 | 0.3117±0.0440 | 0.2564±0.0551 | 0.9052±0.0200 | 0.4189±0.0413 |
| | Exathlon | 25 | 20.16 | 0.7769±0.0248 | 0.9545±0.0072 | 0.6045±0.0287 | 0.7482±0.0257 | 0.9518±0.0075 | 0.8357±0.0159 |
| | GECCO | 1 | 9 | 0.1601±0.0354 | 0.8727±0.0256 | 0.2268±0.0392 | 0.2192±0.0439 | 0.9050±0.0183 | 0.2959±0.0466 |
| | GHL | 23 | 19 | 0.0085±0.0001 | 0.3391±0.0098 | 0.0240±0.0082 | 0.0073±0.0001 | 0.3090±0.0103 | 0.0241±0.0003 |
| | Genesis | 1 | 18 | 0.2964±0.1583 | 0.9902±0.0049 | 0.2930±0.1160 | 0.1945±0.1794 | 0.9867±0.0061 | 0.3023±0.1518 |
| | LTDB | 4 | 2.25 | 0.6066±0.0286 | 0.8640±0.0069 | 0.5965±0.0209 | 0.5856±0.0270 | 0.8327±0.0076 | 0.5861±0.0148 |
| | MITDB | 11 | 2 | 0.3841±0.0113 | 0.8803±0.0042 | 0.4647±0.0100 | 0.4264±0.0151 | 0.8567±0.0046 | 0.4864±0.0106 |
| | MSL | 14 | 55 | 0.2416±0.0400 | 0.7768±0.0240 | 0.3630±0.0402 | 0.1969±0.0427 | 0.7498±0.0280 | 0.2937±0.0346 |
| | OPPORTUNITY | 7 | 248 | 0.1701±0.0289 | 0.6069±0.0387 | 0.3463±0.0277 | 0.1646±0.0310 | 0.5810±0.0371 | 0.2134±0.0293 |
| | PSM | 1 | 25 | 0.2033±0.0128 | 0.6597±0.0155 | 0.2749±0.0126 | 0.1839±0.0131 | 0.6571±0.0179 | 0.2621±0.0059 |
| | SMAP | 25 | 25 | 0.5363±0.0127 | 0.9009±0.0086 | 0.5860±0.0181 | 0.5220±0.0139 | 0.9055±0.0079 | 0.5428±0.0152 |
| | SMD | 20 | 38 | 0.3545±0.0179 | 0.8479±0.0084 | 0.4044±0.0169 | 0.3850±0.0193 | 0.8728±0.0085 | 0.4445±0.0176 |
| | SVDB | 28 | 2 | 0.5574±0.0103 | 0.9050±0.0036 | 0.5436±0.0097 | 0.5259±0.0090 | 0.8814±0.0036 | 0.5496±0.0069 |
| | SWaT | 2 | 58.5 | 0.2428±0.0227 | 0.6864±0.0268 | 0.3421±0.0314 | 0.2174±0.0261 | 0.6679±0.0280 | 0.3213±0.0195 |
| | TAO | 11 | 3 | 0.7273±0.0053 | 0.8535±0.0018 | 0.1848±0.0399 | 0.0849±0.0016 | 0.4966±0.0064 | 0.1591±0.0006 |
| **TSB-AD-M Average** | | | | **0.4263±0.0051** | **0.7940±0.0023** | **0.4065±0.0067** | **0.3772±0.0044** | **0.7623±0.0022** | **0.4275±0.0035** |

Figures 6 and 7 compare the average run times of the methods on TSB-AD-U and TSB-AD-M. PaAno exhibited competitive run time, averaging 5.3712 seconds on TSB-AD-U and 9.6596 seconds on TSB-AD-M. PaAno achieved faster run time than all of the Transformer-based methods, demonstrating superior computational efficiency while maintaining competitive performance. Statistical and machine learning methods generally required less run time, demonstrating their utility for low-latency or resource-constrained scenarios.

Compared to baseline methods, PaAno's advantage in run time becomes more pronounced on TSB-AD-M. While the run time of baseline methods typically increases with the number of channels in the time series, the run time of PaAno primarily depends on retrieving the nearest patch embeddings from the memory bank and is thus highly robust to the number of channels.

### E.4 SCORE DISTRIBUTION

Figure 8 presents the distribution of VUS-PR scores for each method across all time series in the TSB-AD Eval set. The methods are ordered from left to right based on their average VUS-PR scores, and both the median and mean values are visualized to reflect central tendency and consistency. Among them, PaAno exhibited the highest median and mean VUS-PR scores in both univariate and multivariate time-series anomaly detection. The gap between the median and mean was also the smallest, indicating that PaAno achieved a stable performance distribution.

Table 16: Performance comparison of 40 univariate time-series anomaly detection methods on TSB-AD-U.

| Category | Method | VUS-PR | Rank | VUS-ROC | Rank | Range-F1 | Rank | AUC-PR | Rank | AUC-ROC | Rank | Point-F1 | Rank |
|---|---|---|---|---|---|---|---|---|---|---|---|---|---|
| Statistical & Machine Learning | (Sub)-PCA | 0.42 | 2 | 0.76 | 10 | 0.41 | 3 | 0.37 | 3 | 0.71 | 11 | 0.42 | 3 |
| | KShapeAD | 0.40 | 3 | 0.76 | 10 | 0.40 | 4 | 0.35 | 4 | 0.74 | 5 | 0.39 | 4 |
| | POLY | 0.39 | 5 | 0.76 | 10 | 0.35 | 9 | 0.31 | 10 | 0.73 | 8 | 0.37 | 8 |
| | Series2Graph | 0.39 | 5 | 0.80 | 4 | 0.35 | 9 | 0.33 | 5 | 0.76 | 3 | 0.38 | 5 |
| | KMeansAD | 0.37 | 9 | 0.76 | 10 | 0.38 | 6 | 0.32 | 7 | 0.74 | 5 | 0.37 | 8 |
| | MatrixProfile | 0.35 | 11 | 0.76 | 10 | 0.32 | 17 | 0.26 | 20 | 0.73 | 8 | 0.33 | 18 |
| | (Sub)-KNN | 0.35 | 11 | 0.79 | 5 | 0.32 | 17 | 0.27 | 18 | 0.76 | 3 | 0.34 | 16 |
| | SAND | 0.34 | 13 | 0.76 | 10 | 0.36 | 7 | 0.29 | 14 | 0.73 | 8 | 0.35 | 12 |
| | SR | 0.32 | 16 | 0.81 | 3 | 0.35 | 9 | 0.32 | 7 | 0.74 | 5 | 0.38 | 5 |
| | IForest | 0.30 | 19 | 0.78 | 7 | 0.30 | 21 | 0.29 | 14 | 0.71 | 11 | 0.35 | 12 |
| | DLinear | 0.27 | 23 | 0.76 | 10 | 0.24 | 28 | 0.23 | 23 | 0.67 | 19 | 0.28 | 23 |
| | NLinear | 0.26 | 26 | 0.74 | 21 | 0.21 | 34 | 0.20 | 26 | 0.63 | 29 | 0.24 | 28 |
| | (Sub)-LOF | 0.25 | 31 | 0.73 | 23 | 0.25 | 25 | 0.16 | 32 | 0.68 | 16 | 0.24 | 28 |
| | (Sub)-MCD | 0.24 | 32 | 0.72 | 27 | 0.24 | 28 | 0.15 | 36 | 0.67 | 19 | 0.23 | 31 |
| | (Sub)-HBOS | 0.23 | 34 | 0.67 | 38 | 0.27 | 24 | 0.18 | 29 | 0.61 | 32 | 0.23 | 31 |
| | (Sub)-OCSVM | 0.23 | 34 | 0.73 | 23 | 0.23 | 30 | 0.16 | 32 | 0.65 | 25 | 0.22 | 34 |
| | (Sub)-IForest | 0.22 | 36 | 0.72 | 27 | 0.23 | 30 | 0.16 | 32 | 0.63 | 29 | 0.22 | 34 |
| | LOF | 0.17 | 38 | 0.68 | 35 | 0.22 | 32 | 0.14 | 37 | 0.58 | 36 | 0.21 | 37 |
| Conventional Neural Network | KAN-AD | 0.40 | 3 | 0.82 | 2 | 0.43 | 2 | 0.41 | 2 | 0.80 | 2 | 0.44 | 2 |
| | USAD | 0.36 | 10 | 0.71 | 32 | 0.40 | 4 | 0.32 | 7 | 0.66 | 23 | 0.37 | 8 |
| | DeepAnT | 0.34 | 13 | 0.79 | 5 | 0.35 | 9 | 0.33 | 5 | 0.71 | 11 | 0.38 | 5 |
| | LSTMAD | 0.33 | 15 | 0.76 | 10 | 0.34 | 14 | 0.31 | 10 | 0.68 | 16 | 0.37 | 8 |
| | DADA | 0.31 | 18 | 0.77 | 8 | 0.31 | 19 | 0.29 | 14 | 0.71 | 11 | 0.33 | 18 |
| | OmniAnomaly | 0.29 | 22 | 0.72 | 27 | 0.29 | 22 | 0.27 | 18 | 0.65 | 25 | 0.31 | 21 |
| | AutoEncoder | 0.26 | 26 | 0.69 | 34 | 0.28 | 23 | 0.19 | 28 | 0.63 | 29 | 0.25 | 26 |
| | FITS | 0.26 | 26 | 0.73 | 23 | 0.20 | 36 | 0.17 | 31 | 0.61 | 32 | 0.23 | 31 |
| | TimesNet | 0.26 | 26 | 0.72 | 27 | 0.21 | 34 | 0.18 | 29 | 0.61 | 32 | 0.24 | 28 |
| | TranAD | 0.26 | 26 | 0.68 | 35 | 0.25 | 25 | 0.20 | 26 | 0.57 | 37 | 0.25 | 26 |
| | Donut | 0.20 | 37 | 0.68 | 35 | 0.20 | 36 | 0.14 | 37 | 0.56 | 38 | 0.20 | 38 |
| Transformer | MOMENT (FT) | 0.39 | 5 | 0.76 | 10 | 0.35 | 9 | 0.30 | 12 | 0.69 | 15 | 0.35 | 12 |
| | MOMENT (ZS) | 0.38 | 8 | 0.75 | 20 | 0.36 | 7 | 0.30 | 12 | 0.68 | 16 | 0.35 | 12 |
| | iTransformer | 0.32 | 16 | 0.77 | 8 | 0.22 | 32 | 0.21 | 25 | 0.67 | 19 | 0.26 | 25 |
| | PatchTST | 0.30 | 19 | 0.76 | 10 | 0.25 | 25 | 0.23 | 23 | 0.65 | 25 | 0.27 | 24 |
| | TimesFM | 0.30 | 19 | 0.74 | 21 | 0.34 | 14 | 0.28 | 17 | 0.67 | 19 | 0.34 | 16 |
| | Chronos | 0.27 | 23 | 0.73 | 23 | 0.33 | 16 | 0.26 | 20 | 0.66 | 23 | 0.32 | 20 |
| | Lag-Llama | 0.27 | 23 | 0.72 | 27 | 0.31 | 19 | 0.25 | 22 | 0.65 | 25 | 0.30 | 22 |
| | OFA | 0.24 | 32 | 0.71 | 32 | 0.20 | 36 | 0.16 | 32 | 0.59 | 35 | 0.22 | 34 |
| | AnomalyTransformer | 0.12 | 39 | 0.56 | 40 | 0.14 | 39 | 0.08 | 39 | 0.50 | 39 | 0.12 | 39 |
| | DCdetector | 0.09 | 40 | 0.57 | 39 | 0.07 | 40 | 0.05 | 40 | 0.50 | 40 | 0.10 | 40 |
| Ours | PaAno | 0.53 | 1 | 0.89 | 1 | 0.49 | 1 | 0.47 | 1 | 0.87 | 1 | 0.52 | 1 |

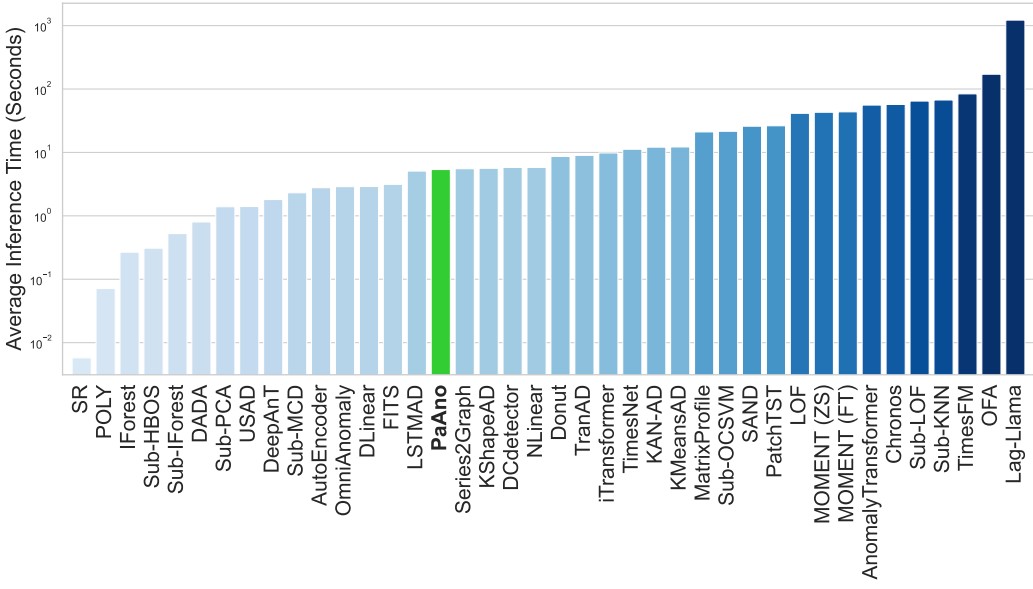

Figure 6: Average run time on TSB-AD-U.

Table 17: Performance comparison of 32 multivariate time-series anomaly detection methods on TSB-AD-M.

| Category | Method | VUS-PR | Rank | VUS-ROC | Rank | Range-F1 | Rank | AUC-PR | Rank | AUC-ROC | Rank | Point-F1 | Rank |
|---|---|---|---|---|---|---|---|---|---|---|---|---|---|
| Statistical & Machine Learning | PCA | 0.31 | 3 | 0.74 | 4 | 0.29 | 11 | 0.31 | 4 | 0.70 | 4 | 0.37 | 3 |
| | NLinear | 0.30 | 8 | 0.70 | 13 | 0.21 | 19 | 0.26 | 13 | 0.65 | 14 | 0.31 | 13 |
| | DLinear | 0.29 | 14 | 0.70 | 13 | 0.21 | 19 | 0.27 | 10 | 0.66 | 12 | 0.32 | 10 |
| | KMeansAD | 0.29 | 14 | 0.73 | 6 | 0.33 | 7 | 0.25 | 15 | 0.69 | 6 | 0.31 | 13 |
| | CBLOF | 0.27 | 16 | 0.70 | 13 | 0.31 | 9 | 0.28 | 8 | 0.67 | 8 | 0.32 | 10 |
| | MCD | 0.27 | 16 | 0.69 | 16 | 0.20 | 24 | 0.27 | 10 | 0.65 | 14 | 0.33 | 8 |
| | OCSVM | 0.26 | 18 | 0.67 | 22 | 0.30 | 10 | 0.23 | 18 | 0.61 | 23 | 0.28 | 19 |
| | RobustPCA | 0.24 | 20 | 0.61 | 28 | 0.33 | 7 | 0.24 | 16 | 0.58 | 25 | 0.29 | 16 |
| | EIF | 0.21 | 21 | 0.71 | 9 | 0.26 | 13 | 0.19 | 22 | 0.67 | 8 | 0.26 | 22 |
| | COPOD | 0.20 | 24 | 0.69 | 16 | 0.24 | 15 | 0.20 | 20 | 0.65 | 14 | 0.27 | 21 |
| | IForest | 0.20 | 24 | 0.69 | 16 | 0.24 | 15 | 0.19 | 22 | 0.66 | 12 | 0.26 | 22 |
| | HBOS | 0.19 | 26 | 0.67 | 22 | 0.24 | 15 | 0.16 | 24 | 0.63 | 22 | 0.24 | 24 |
| | KNN | 0.18 | 28 | 0.59 | 30 | 0.21 | 19 | 0.14 | 27 | 0.51 | 31 | 0.19 | 29 |
| | LOF | 0.14 | 30 | 0.60 | 29 | 0.14 | 30 | 0.10 | 30 | 0.53 | 29 | 0.15 | 30 |
| Conventional Neural Network | KAN-AD | 0.40 | 2 | 0.76 | 2 | 0.41 | 1 | 0.35 | 2 | 0.73 | 2 | 0.40 | 2 |
| | DADA | 0.31 | 3 | 0.73 | 6 | 0.25 | 14 | 0.31 | 4 | 0.69 | 6 | 0.35 | 6 |
| | DeepAnT | 0.31 | 3 | 0.76 | 2 | 0.37 | 4 | 0.32 | 3 | 0.73 | 2 | 0.37 | 3 |
| | LSTMAD | 0.31 | 3 | 0.74 | 4 | 0.38 | 3 | 0.31 | 4 | 0.70 | 4 | 0.36 | 5 |
| | OmniAnomaly | 0.31 | 3 | 0.69 | 16 | 0.37 | 4 | 0.27 | 10 | 0.65 | 14 | 0.32 | 10 |
| | AutoEncoder | 0.30 | 8 | 0.69 | 16 | 0.28 | 12 | 0.30 | 7 | 0.67 | 8 | 0.34 | 7 |
| | USAD | 0.30 | 8 | 0.68 | 21 | 0.37 | 4 | 0.26 | 13 | 0.64 | 19 | 0.31 | 13 |
| | Donut | 0.26 | 18 | 0.71 | 9 | 0.21 | 19 | 0.20 | 20 | 0.64 | 19 | 0.28 | 19 |
| | FITS | 0.21 | 21 | 0.66 | 24 | 0.16 | 29 | 0.15 | 25 | 0.58 | 25 | 0.22 | 25 |
| | TimesNet | 0.19 | 26 | 0.64 | 26 | 0.17 | 27 | 0.13 | 29 | 0.56 | 27 | 0.20 | 28 |
| | TranAD | 0.18 | 28 | 0.65 | 25 | 0.21 | 19 | 0.14 | 27 | 0.59 | 27 | 0.21 | 26 |
| Transformer | CATCH | 0.30 | 8 | 0.72 | 8 | 0.18 | 25 | 0.23 | 18 | 0.65 | 14 | 0.29 | 16 |
| | iTransformer | 0.30 | 8 | 0.71 | 9 | 0.18 | 25 | 0.24 | 16 | 0.64 | 19 | 0.29 | 16 |
| | PatchTST | 0.30 | 8 | 0.71 | 9 | 0.22 | 18 | 0.28 | 8 | 0.67 | 8 | 0.33 | 8 |
| | OFA | 0.21 | 21 | 0.63 | 27 | 0.17 | 27 | 0.15 | 25 | 0.55 | 28 | 0.21 | 26 |
| | AnomalyTransformer | 0.12 | 31 | 0.57 | 31 | 0.14 | 30 | 0.07 | 31 | 0.52 | 30 | 0.12 | 31 |
| | DCdetector | 0.10 | 32 | 0.54 | 32 | 0.09 | 32 | 0.05 | 32 | 0.48 | 32 | 0.10 | 32 |
| Ours | PaAno | 0.43 | 1 | 0.79 | 1 | 0.41 | 1 | 0.38 | 1 | 0.76 | 1 | 0.43 | 1 |

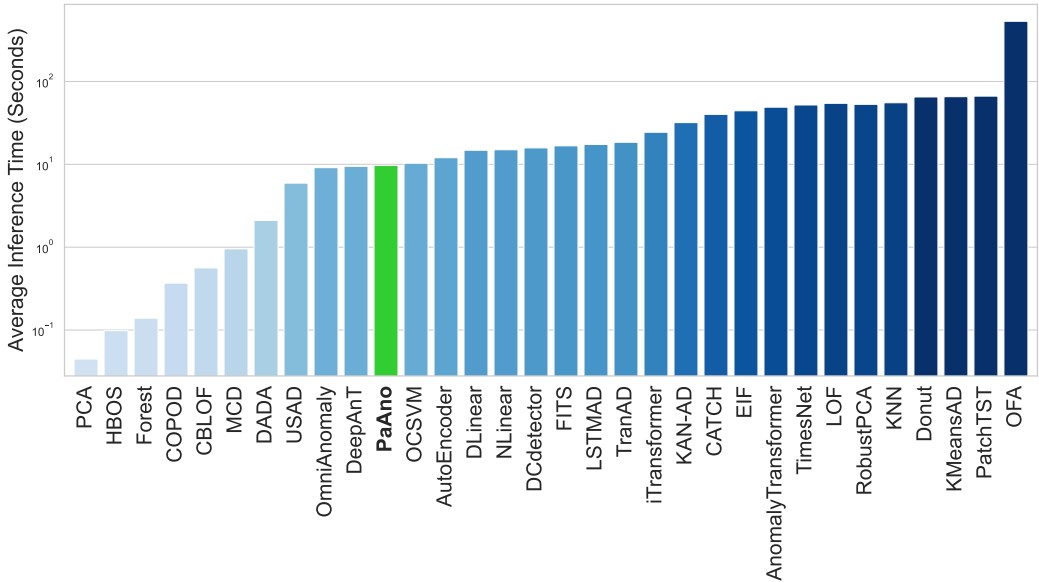

Figure 7: Average run time on TSB-AD-M.

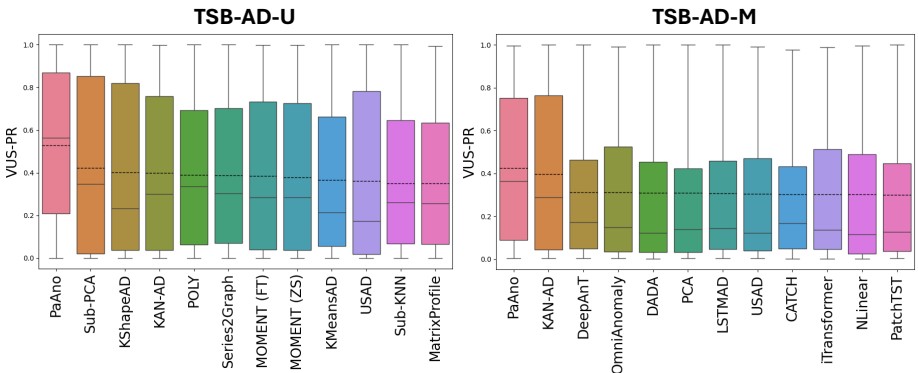

Figure 8: Boxplot of VUS-PR distributions for TSB-AD-U and TSB-AD-M. The dashed line and solid line represent the mean and median values, respectively. Only the top-12 methods ranked by average VUS-PR are presented, ordered from left to right accordingly.

