# OpenReview forum: "PaAno: Patch-Based Representation Learning for Time-Series Anomaly Detection"
_ICLR.cc/2026/Conference — ICLR 2026 Poster_

### Official Review · Reviewer_EcX6 · 2025-10-29

**Soundness:** 3
**Presentation:** 4
**Contribution:** 4
**Rating:** 6
**Confidence:** 3

**Summary:**

This paper proposes PaAno (Patch-based representation learning for time-series Anomaly detection), a lightweight framework for semi-supervised time-series anomaly detection. Instead of using large transformer or foundation models, PaAno applies a 1D-CNN to extract vector embeddings from short overlapping temporal patches. The model is trained with a triplet loss (for metric learning) and a pretext loss (for predicting temporal adjacency between patches). A memory bank of normal patch embeddings is built for inference, where anomalies are identified via distance to nearest normal embeddings. Evaluated on the TSB-AD benchmark, PaAno achieves state-of-the-art results across univariate and multivariate datasets, outperforming heavier architectures while remaining computationally efficient. Ablation and sensitivity analyses confirm the contributions of both loss terms and robustness to hyperparameters.

**Strengths:**

Originality: Fair

Quality: Good

Clarity: Fair

Significance: Good


Additional note: The paper is mostly well written and the results are extremely promising. The methodology is clearly presented and mostly easy to follow. The methodology is an extension of existing research but is applied to a new domain in a clever and relevant way.

**Weaknesses:**

Hyper-parameter tuning: The paper claims the model is resilient to hyper-param tuning, but the paper does not show the affect of tuning the hyper params in triplet loss and combined loss. Additionally, as the triplet loss samples the current mini batch at inference time, would the mini batch size not be a big hyper-param design consideration?
Training details: There is no motivation given to as why the pretext loss weight decays to 0 within the first 20 iterations, this is in the appendix, I believe this is a major detail that needs to be discussed as the paper claims that the pretext loss is essential for the model performance.

Generalization to other datasets: Only one dataset was used to evaluate methods.

Formatting/Editing Mistakes: Table 10 in Appendix E3 shows a row called Total , but it shows (what i assume is) the mean of the different metrics in the table.

Limited theoretical grounding: The paper lacks formal analysis explaining why the patch-based embedding space generalizes effectively to unseen anomalies.

Unsubstantiated assertions: In section 3.1, the paper asserts that “Modelling long sequences with heavy global attention can dilute these local temporal dependencies”. This is not backed by a citation or theoretical or empirical proof. Additionally the term “heavy” here is ambiguous.

**Questions:**

What is the motivation behind decaying the weight for pretext loss?

In section 3.5, a visual representation of even a pseudo code of the anomaly detection method would aid in clarity.

Is the sampling of negative patch done at inference time during the forward loop?

What is the embedding space that is used to select a negative sample? Is it the embedding space of the mode? If so how is the initialization handled when the model is not trained?

In section 3,3, i need some clarification. I think there is an error. THe paper states “ We define the negative patch pi as the hard negative, chosen as the patch in the minibatch B that has the smallest cosine distance to pi in the embedding space, i.e.,
pj ∈ B \ {pi } that minimizes dist(fθ (pi ), fθ (pj )).“ Shouldnt the distance be maximized here to find a negative patch that is the most dissimilar to the patch p?

I am willing to improve the rating if my questions are answered.

---

> ### Author Response · Authors · 2025-11-17
>
> Dear Reviewer EcX6,
>
> We sincerely thank you for your thoughtful and constructive questions. Your comments were highly insightful and allowed us to clarify several important design choices in PaAno. Below, we provide detailed responses to each of your questions.
>
> **W1&Q1. Motivation behind decaying the weight for pretext loss**
>
> The pretext loss was introduced to help stabilize the encoder during the early iterations of training, a phase in which the embedding space is unstable. Because the triplet loss depends
> on relative distances within each minibatch, it can produce noisy or unreliable updates when the embedding space has not yet formed. Incorporating the pretext objective during this stage improves training stability, and we observed consistent gains in performance when it is used in the early phase.
> However, keeping the pretext loss active throughout all iterations was empirically detrimental. Once the embedding space becomes more structured, the pretext objective begins to interfere with the triplet loss, leading to degraded performance. This motivated us to limit the pretext loss to the early part of training.
>
>  We also found that abruptly removing the pretext loss can destabilize the training process. To avoid this, we adopted a linear-decay schedule that gradually reduces its weight. Across all tested ratios, the linear-decay strategy consistently outperformed the constant schedule, indicating that a smooth reduction leads to more stable and effective training behavior. ("pretext 10%" means that the pretext loss is applied only during the first 10% of the training iterations.) This rationale has been added to the revised paper.
>
>
> **TSB-AD-U**
>
> | Setting           | VUS-PR | VUS-ROC | Range-F1 | AUC-PR | AUC-ROC | Point-F1 |
> |-------------------|--------|---------|----------|--------|---------|-----------|
> | pretext 10%, linear decayed   | 0.5188 | 0.8863  | 0.4819 | 0.4559 | 0.8573 |	0.5068 |
> | pretext 10%, constant  | 0.4998 | 0.8811  | 0.4743   | 0.4412 | 0.8529  | 0.4934    |
> | pretext 50%, linear decayed   | 0.4864 | 0.8749  | 0.4661   | 0.4220 | 0.8452  | 0.4700   |
> | pretext 50%, constant  | 0.4729 | 0.8763  | 0.4614   | 0.4129 | 0.8455  | 0.4618    |
> | pretext 100%, linear decayed    | 0.4683 | 0.8712  | 0.4531   | 0.4043 | 0.8401  | 0.4582   |
> | pretext 100%, constant  | 0.4719 | 0.8735  | 0.4551   | 0.4081 | 0.8430  | 0.4591    |
>
> **TSB-AD-M**
>
> | Setting           | VUS-PR | VUS-ROC | Range-F1 | AUC-PR | AUC-ROC | Point-F1 |
> |-------------------|--------|---------|----------|--------|---------|-----------|
> | pretext 10%, linear decayed   | 0.4307 | 0.7928 | 0.4141 | 0.3830 | 0.7613 |	0.4299 |
> | pretext 10%, constant  | 0.4301 | 0.7912  | 0.4062   | 0.3811 | 0.7601  | 0.4290    |
> | pretext 50%, linear decayed   | 0.4010 | 0.7791  | 0.3980   | 0.3582 | 0.7481  | 0.4100   |
> | pretext 50%, constant  | 0.3942 | 0.7721  | 0.3883   | 0.3502 | 0.7402  | 0.4011    |
> | pretext 100%, linear decayed    | 0.3997 | 0.7662  | 0.3963   | 0.3541 | 0.7312  | 0.3998   |
> | pretext 100%, constant  | 0.3916 | 0.7669  | 0.3832   | 0.3492 | 0.7351  | 0.3980    |
>
> **Q2. Request for pseudo-code of the 3.5 Anomaly Detection of PaAno.**
>
> We have added the pseudocode for Section 3.5 (Anomaly Detection). Thank you very much for the insightful suggestion. We will upload the revised manuscript as soon as the remaining experiments are completed.
>
> **Q3. Is the sampling of negative patch done at inference time during the forward loop?**
>
> During inference, the anomaly score is computed using the encoder $f_\theta$ and the reduced memory bank $\hat{\mathcal{M}}$, without any negative patch sampling. The patch-level anomaly score is defined as:
>
>
> $S(p_t) = \frac{1}{k} \sum_{i=1}^{k} \text{dist}( f_\theta(p_t), m_t^{(i)} )$
>
>
>
> where $\mathbf{m}_t^{(i)}$ denotes the $k$ nearest neighbors retrieved from the reduced memory bank $\hat{\mathcal{M}}$.
>
>
> **Q4. What embedding space is used for negative selection, and how is it initialized before training?**
>
> All negatives are selected in the current embedding space produced by the encoder $f_\theta$. At the start of training, the embedding space is defined by the encoder’s random (default PyTorch) initialization, and it becomes structured as the model trains.

---

> ### Author Response · Authors · 2025-11-17
>
> **Q5.  Why is the negative patch defined as the closest non-positive sample rather than the most dissimilar one?**
>
> Originally, the negative patch in PaAno was intentionally chosen as the closest non-positive patch in the minibatch rather than the most distant one. (**However, PaAno is now updated to use the farthest negative as default, thanks to valuable discussions with you.**)
>
> We thought the closest negative would offer informative contrast for refining ambiguous regions of the representation space[1,2]. We assigned exactly one hard negative per anchor for this reason. Each anchor also has a single positive patch, defined as a slightly shifted version of the anchor.
> Under this setting, the positive encourages the encoder to remain robust to small temporal shifts, while the closest non-positive patch promotes discrimination between patterns that are visually similar but not temporally adjacent. Metric learning with hard negatives is empirically validated through comparisons against InfoNCE loss, achieving superior performance on both TSB-AD-U and TSB-AD-M.
>
>
> **TSB-AD-U**
>
> | Setting           | VUS-PR | VUS-ROC | Range-F1 | AUC-PR | AUC-ROC | Point-F1 |
> |-------------------|--------|---------|----------|--------|---------|-----------|
> | PaAno (Hard Negative ver)    | 0.5085 | 0.8816 | 0.4770 |	0.4494 | 0.8536 | 0.5001 |
> | PaAno (InfoNCE)   | 0.4835 | 0.8670  | 0.4528   | 0.4264 | 0.8362  | 0.4775    |
>
> **TSB-AD-M**
>
> | Setting           | VUS-PR | VUS-ROC | Range-F1 | AUC-PR | AUC-ROC | Point-F1 |
> |-------------------|--------|---------|----------|--------|---------|-----------|
> | PaAno (Hard Negative ver)    | 0.4102 | 0.7857| 0.3921| 0.3671| 0.7539 | 0.4130 |
> | PaAno (InfoNCE)   | 0.3704 | 0.7619  | 0.3539   | 0.3281 | 0.7284  | 0.3721    |
>
>
> **W1: Necessity of Sensitivity Analysis about loss weight and mini batch size**
>
> We fully agree that understanding the sensitivity of the hyperparameters, including the loss weight and the minibatch size, is important for evaluating the robustness of PaAno. To address this, we conducted additional sensitivity experiments on both loss weight and the batch size, and the results are presented below. We will add this experiment in the revised version of the paper as well!
>
>
>
> **loss weight(TSB-AD-U)**
> | lambda          | VUS-PR | VUS-ROC | Range-F1 | AUC-PR | AUC-ROC | Point-F1 |
> |-------------------|--------|---------|----------|--------|---------|-----------|
> | 0.1 | 0.5146 | 0.8872  | 0.4811  | 0.4510 | 0.8592  | 0.5043    |
> | 0.5 | 0.5091 | 0.8840  | 0.4702   | 0.4491 | 0.8555  | 0.5001    |
> | 1.0 | 0.5188 | 0.8863 | 0.4819 | 0.4559 | 0.8573 | 0.5068   |
> | 2.0 | 0.5102 | 0.8854  | 0.4722   | 0.4510 | 0.8563  | 0.5011    |
>
> **loss weight(TSB-AD-M)**
> | lambda          | VUS-PR | VUS-ROC | Range-F1 | AUC-PR | AUC-ROC | Point-F1 |
> |-------------------|--------|---------|----------|--------|---------|-----------|
> | 0.1 | 0.4344 | 0.7991  | 0.4200 | 0.3852 | 0.7673  | 0.4324    |
> | 0.5 | 0.4309 | 0.7919  | 0.4102   | 0.3864 | 0.7601  | 0.4311   |
> | 1.0 | 0.4307 | 0.7928 | 0.4141 | 0.3830 | 0.7613 | 0.4299 |
> | 2.0 | 0.4299 | 0.7925  | 0.4072   | 0.3851 | 0.7611 | 0.4312   |
>
> **Batch size(TSB-AD-U)**
> | Batch size         | VUS-PR | VUS-ROC | Range-F1 | AUC-PR | AUC-ROC | Point-F1 |
> |-------------------|--------|---------|----------|--------|---------|-----------|
> | 128 | 0.5079 | 0.8798  | 0.4653 | 0.4476 | 0.8494  | 0.4997    |
> | 256 | 0.5108 | 0.8844  | 0.4706   | 0.4524 | 0.8546  | 0.5021    |
> | 512 | 0.5188 | 0.8863 | 0.4819 | 0.4559 | 0.8573 | 0.5068   |
> | 1024 | 0.5084 | 0.8821  | 0.4682   | 0.4481 | 0.8528  | 0.4991    |
>
> **Batch size(TSB-AD-M)**
> | Batch size         | VUS-PR | VUS-ROC | Range-F1 | AUC-PR | AUC-ROC | Point-F1 |
> |-------------------|--------|---------|----------|--------|---------|-----------|
> | 128 | 0.4231 | 0.7903  | 0.3991  | 0.3782 | 0.7592  | 0.4232    |
> | 256 | 0.4271 | 0.7930  | 0.4052   | 0.3811 | 0.7608  | 0.4280    |
> | 512 | 0.4307 | 0.7928 | 0.4141 | 0.3830 | 0.7613 | 0.4299 |
> | 1024 | 0.4288 | 0.7948  | 0.4084   | 0.3806 | 0.7631  | 0.4270    |
>
> **W2: Necessity of Multiple Dataset Evaluation**
>
> The TSB-AD benchmark consists of 40 well-curated datasets for anomaly detection, each of which contains multiple time series [3]. As summarized in Table 1, TSB-AD is further divided into two subsets: a set of time-series explicitly reserved for hyperparameter tuning (Tuning set) and a substantially larger set used for performance evaluation of certain methods (Eval set). Each time-series has a predefined train/test split. We provided dataset-wise results in Appendix E.1, which report the performance across the 530 time-series in TSB-AD-U and the 180 time-series in TSB-AD-M.
>
> [1] Robinson et al. Contrastive Learning with Hard Negative Samples, ICLR 2021.
>
> [2] Xuan et al. Hard Negative examples are hard, but useful. ECCV 2020.
>
> [3] Liu et al. The Elephant in the Room: Towards A Reliable Time-Series Anomaly Detection Benchmark, Neurips 2025.

---

> ### Author Response · Authors · 2025-11-17
>
> **W3: Meaning of "Total" in Table 10 of Appendix E3**
>
> The row labeled “Total” indicates the mean performance computed over all subsets within TSB-AD-U or TSB-AD-M, rather than an additional subset. We apologize for the confusion and will consider revising the table presentation to make this clearer.
>
> **W5: Citation for claims about transformers diluting local temporal patterns**
>
> Transformers effectively capture long-range dependencies through global self-attention, but this global focus can dilute fine-grained local dynamics that are crucial in time-series data. Prior work has shown that, despite strong long-term modeling ability, transformer-based models often struggle to represent localized temporal patterns [4,5,6].  We revised our paper and made the citation accordingly. Also we removed “heavy” in that sentence. Thank you for your insightful comments!
>
>
> [4] Li et al. SMARTformer: Semi-Autoregressive Transformer with Efficient Integrated Window Attention for Long Time Series Forecasting, IJCAI 2023.
>
> [5] Li et al. Enhancing the Locality and Breaking the Memory Bottleneck of Transformer on Time Series Forecasting, NeurIPS 2019.
>
> [6] Oliveira et al. Evaluating the Effectiveness of Time Series Transformers for Demand Forecasting in Retail, Mathematics 2024.

---

> ### Author Response · Authors · 2025-11-26
> **Waiting for response**
>
> Dear Reviewer EcX6,
>
> As the rebuttal period is coming to a close, we would like to kindly ask whether our responses have adequately addressed your primary concerns. We believe we have provided answers to all the questions you raised. We would be grateful if you could kindly review them, and if any further points require clarification or discussion, we would warmly welcome the opportunity to continue the conversation.
>
> If our responses sufficiently address your concerns, we would also be sincerely grateful if you could consider updating your score accordingly.
>
> We truly appreciate you taking the time to review our work and provide thoughtful feedback, especially amidst your busy schedule.
>
> Thank you once again for your valuable time and constructive comments.
>
> Kind regards,
>
>
> The Authors

---

> > ### Comment · Reviewer_EcX6 · 2025-11-26
> > **Response to Rebuttal**
> >
> > Dear authors, thank you for your responses, i do believe it improves your submission and i would rate it 7 now. I can only choose 6 or 8 as the rating and i believe it is still closer to a 6 than an 8.
> > Regards

---

> > > ### Author Response · Authors · 2025-11-26
> > >
> > > Dear Reviewer EcX6,
> > >
> > > Thank you very much for your kind follow-up and for reconsidering our manuscript. After a careful review of our previous responses, we have identified aspects that were not sufficiently addressed. We are now working to further clarify the remaining points raised in your review. We will soon provide an additional response.
> > >
> > > Kind regards,
> > >
> > > The Authors

---

> ### Author Response · Authors · 2025-11-29
>
> Dear Reviewer EcX6,
>
> We have carefully reflected on the remaining points you raised, and we would like to supplement our earlier responses.
> We sincerely thank you for your insightful questions. All corresponding clarifications will be incorporated into the final revised manuscript.
>
> **Q2. Request for pseudo-code of the 3.5 Anomaly Detection of PaAno.**
>
> To clarify the understanding of the anomaly detection procedure in PaAno, we added a new figure (Figure 4) illustrating the step-by-step process as well as its pseudo code. We also added the explanation of them in the main paper as well!
>
> **Q4 : What embedding space is used for negative selection, and how is it initialized before training?**
>
> The encoder is initially randomly initialized (standard PyTorch defaults), causing the embedding space at the beginning to lack meaningful structure. Thus, selecting the most dissimilar patch at the start of training would be similar to a quasi-random choice.
>
> However, as training progresses, the embedding space becomes increasingly structured. Also, the pretext task in the initial training iteration helps stabilize this early unstable phase of triplet loss-based training. These procedures allow the negative selection to progressively capture meaningful dissimilarities, making it an effective contrastive sample.
>
> **Q5 : Why is the negative patch defined as the closest non-positive sample rather than the most dissimilar one?**
>
> To further analyze the negative selection in triplet formulation in PaAno, we conducted a thorough sensitivity analysis of the negative sampling strategy. We evaluated the farthest patch, the closest patch, a randomly selected patch, and patches chosen at fixed percentiles of the distance ranking (50%) in the embedding space.
>
>
>
>
> **Triplet loss compared with InfoNCE and various negative selection strategy**
>
> (**Bold** = best, (Parentheses) = second best (all values in %))
>
> | Dataset | Loss | Negative Sampling | VUS-PR | VUS-ROC | Range-F1 | AUC-PR | AUC-ROC | Point-F1 |
> |--------|------|------------------|--------|---------|-----------|---------|----------|-----------|
> | **TSB-AD-U** | Triplet | Random | 49.6 | 87.2 | 47.0 | 44.5 | 84.3 | 49.2 |
> | | Triplet | (Closest) | (50.9) | (88.2) | (47.7) | (45.0) | (85.4) | (50.0) |
> | | Triplet | Top 50% | 45.5 | 84.9 | 42.9 | 39.7 | 81.8 | 45.0 |
> | | Triplet | **Farthest** | **51.9** | **88.6** | **48.2** | **45.6** | **85.7** | **50.7** |
> | | InfoNCE | All Non-Self Patch | 48.4 | 86.7 | 45.3 | 42.6 | 83.6 | 47.8 |
>
> | Dataset | Loss | Negative Sampling | VUS-PR | VUS-ROC | Range-F1 | AUC-PR | AUC-ROC | Point-F1 |
> |--------|------|------------------|--------|---------|-----------|---------|----------|-----------|
> | **TSB-AD-M** | Triplet | Random | 36.9 | 76.7 | 35.1 | 32.4 | 73.5 | 37.2 |
> | | Triplet | (Closest) | (41.0) | (78.6) | (39.2) | (36.7) | (75.4) | (41.3) |
> | | Triplet | Top 50% | 37.2 | 76.9 | 35.8 | 32.3 | 74.0 | 37.3 |
> | | Triplet | **Farthest** | **43.1** | **79.3** | **41.4** | **38.3** | **76.1** | **43.0** |
> | | InfoNCE | All Non-Self Patch | 37.0 | 76.2 | 35.4 | 32.8 | 72.8 | 37.2 |
>
>
> Firstly, **The farthest negative** achieved the best performance in both TSB-AD-U and TSB-AD-M. This is because the farthest patch provides a reliably distinct contrast from the anchor, which helps reinforce separation between semantically different normal clusters effectively. Farthest negative encourages the encoder to carve out clearly distinguishable regions in the embedding space, forming clusters that represent truly different normal dynamics.
>
> Nextly, the closest negative ranked second. It still offers a meaningful learning signal, as it forces the encoder to identify subtle differences among highly similar normal patterns. However, compared to the farthest negative, this leads to finer but more fragmented sub-clusters. As a result, the overall normal representation becomes less compact, and anomalies are not as strongly separated as when using the farthest negative.
>
> In contrast, percentile-based negatives and random negatives consistently produced weak contrast and often behaved similarly to InfoNCE. This is because time-series patches obtained from sliding windows have no semantic labels, and patches in the minibatch do not reliably correspond to meaningful negative relations. Effective contrast requires a clear relationship in the embedding space, and only the most similar or the most dissimilar patches reliably satisfy this condition.
>
> Based on these findings, **we updated the method to define the negative as the farthest non-positive patch.** This revision provides a more stable and discriminative contrast for metric learning, and we will incorporate this change into the revised manuscript. Once again, we sincerely appreciate your valuable insight!  It directly contributed to improving both the method and the clarity of the paper.

---

> ### Author Response · Authors · 2025-11-29
>
> **W4: Theoretical grounding for Patch-based embedding space generalizes to unseen anomalies.**
>
> The patch-based embedding space in PaAno can generalize to unseen anomalies because it is organized into well-separated, compact normal representation clusters, causing anomalous patches to lie at distinctly larger distances from any normal region.
>
> The encoder in PaAno is trained via Triplet Loss and a Pretext Loss. The Triplet Loss structures the embedding space by pushing apart the most different normal patch pairs while keeping similar patches (those differing by small temporal shifts) close together. The Pretext Loss complements this process by guiding the encoder to capture temporal adjacency, which helps stabilize representation learning.
>
> Through these objectives, the encoder forms a representation space **where similar normal patches are tightly grouped, while distinct normal patterns occupy different regions of the space.** Consequently, the memory bank used in anomaly detection becomes a collection of cohesive normal clusters that still maintain clear separation between various types of normal behavior.
>
> When an unseen anomaly appears, its representation does not fit into any of these normal clusters. Its distance to surrounding normal embeddings becomes substantially larger than the within-cluster distances for normal patches. As a result, the patch-based embedding space in PaAno can effectively generalize to unseen anomalies.
>
> **W5: Citation for claims about transformers diluting local temporal patterns**
>
> We added the appropriate citations and provided further clarification to support our original claims.
>
> The dilution of local temporal patterns is inherent to the Transformer's global self-attention mechanism, which is locality-agnostic. While this mechanism effectively models long-range dependencies, localized patterns around anomaly points often serve as key cues for time-series anomaly detection [1]. Global attention lacks a local inductive bias and cannot prioritize these essential nearby relationships. Prior studies confirm that this structural trait weakens the fidelity of fine-grained local dynamics, which are crucial for detecting point or contextual anomalies [2, 3, 4].
>
> [1] Yue et al. Sub-Adjacent Transformer: Improving Time Series Anomaly Detection with Reconstruction Error from Sub-Adjacent Neighborhoods, IJCAI 2024.
>
> [2] Li et al. SMARTformer: Semi-Autoregressive Transformer with Efficient Integrated Window Attention for Long Time Series Forecasting, IJCAI 2023.
>
> [3] Li et al. Enhancing the Locality and Breaking the Memory Bottleneck of Transformer on Time Series Forecasting, NeurIPS 2019.
>
> [4] Oliveira et al. Evaluating the Effectiveness of Time Series Transformers for Demand Forecasting in Retail, Mathematics 2024.

---

> ### Author Response · Authors · 2025-11-29
> **Final Remarks and Appreciation**
>
> We will incorporate all of these refinements into the final revised manuscript, and we hope that these additional clarifications address the aspects that were not fully resolved earlier.
>
> We are sincerely grateful for the time and thoughtful attention you have devoted to reviewing our work, especially amidst your busy schedule. It has been a privilege to receive your detailed feedback, and we truly appreciate all your efforts.
>
>
>
> Kind regards,
>
>
> The Authors

---

### Official Review · Reviewer_2Foy · 2025-10-30

**Soundness:** 3
**Presentation:** 3
**Contribution:** 3
**Rating:** 6
**Confidence:** 4

**Summary:**

The paper presents PaAno (Patch-based Anomaly Detection), a lightweight and effective framework for time-series anomaly detection. The core idea is based on patch-based representation learning, where the time series is divided into overlapping temporal patches that are embedded using a convolutional encoder. After training, embeddings of normal patches are stored, and during inference, anomaly scores are computed as the average distance between new patch embeddings and their nearest neighbors in the memory bank

**Strengths:**

* The paper presents time-series anomaly detection (TSAD) method by adopting the TSB-AD benchmark and employing lag-tolerant, threshold-independent metrics (with VUS-PR as the primary measure). It introduces a lightweight patch-encoder combined with metric learning and a memory-bank kNN anomaly scoring approach, applicable to both univariate and multivariate data. The emphasis is on robustness and efficiency rather than architectural complexity.

* Paper combines simple components (1D-CNN, triplet loss, adjacency pretext, memory bank + kNN) yet yields strong empirical gains across multiple range- and point-wise metrics and both U/M benchmarks. The result that such a compact model outperforms many large transformer/foundation approaches under the TSB-AD protocol is interesting and practically relevant.

**Weaknesses:**

* Novelty is incremental. The components are known, the value lies in the clean, effective combination and rigorous evaluation.

* There is little theoretical analysis explaining why triplet + adjacency pretext together produce the observed gains, or conditions when patch locality will fail (e.g., very long-range contextual anomalies). A short analysis or toy case study would strengthen the paper

* All main results use TSB-AD (albeit a rigorous benchmark). It remains unclear how PaAno performs when anomalies are primarily global (long temporal context) or when the training set has non-stationary normals. Clarification on this can further strengthen the paper.

* The ablations show triplet vs. pretext importance and sensitivity to k/memory size,. It would be great to further explore: (a) patch length w effects across dataset types, (b) encoder depth/width tradeoffs, (c) alternative metric losses (contrastive. etc ). These would clarify design choices.

**Questions:**

* Sensitivity beyond reported ranges for patch size, lag tolerance, and neighbor count. Are there regimes where PaAno fails (e.g., nonlocal anomalies or regime shifts)?

* Any leakage from pretext construction across windows, and how is it prevented at train/test boundaries?

* Why was triplet loss chosen over other metric objectives (e.g. Contrastive)? Any stability/convergence comments?

* What are the hardware and exact runtime measurement details (GPU/CPU model, batch sizes during inference, parallelization)? Add them to the appendix or a short paragraph.

* How does PaAno behave under non-stationary normal regimes (drift)? Any mechanisms (or easy extensions) for updating the memory bank online?

---

> ### Author Response · Authors · 2025-11-17
>
> Dear Reviewer 2Foy, we sincerely thank you for providing a detailed review and insightful comments.
>
> **W2,3 & Q1,5: Robustness to Non-Stationarity(regime shifts, drift) and Non-Local Anomalies**
>
> In response to your insightful suggestions, we revisited the normalization step and incorporated **Instance Normalization** as a default component of PaAno to deal with regime shifts and non-stationarity.
>
> Instance Normalization standardizes each input patch to zero mean and unit variance, which stabilizes patch embeddings under gradual level or scale drift [1,2] and prevents non-stationary normals from distorting the memory bank. This modification directly strengthened the performance of PaAno. Across TSB-AD-U and TSB-AD-M, PaAno with Instance Normalization consistently outperformed the original version. Thank you again for your thoughtful feedback, which helped us improve PaAno and strengthen its overall reliability!
>
> **PaAno Performance (with / without InstanceNorm)**
>
> **TSB-AD-U**
>
> | Setting                 | VUS-PR  | VUS-ROC | Range-F1 | AUC-PR | AUC-ROC | Point-F1 |
> |-------------------------|---------|---------|----------|--------|---------|-----------|
> | PaAno (StandardNorm)        | 0.4632  | 0.8502  | 0.4634   | 0.4259 | 0.8253  | 0.4734    |
> | PaAno (InstanceNorm)    | **0.5188**  | **0.8863**  | **0.4819**   | **0.4559** | **0.8573**  | **0.5068**    |
>
> **TSB-AD-M**
>
> | Setting                 | VUS-PR  | VUS-ROC | Range-F1 | AUC-PR | AUC-ROC | Point-F1 |
> |-------------------------|---------|---------|----------|--------|---------|-----------|
> | PaAno (StandardNorm)        | 0.3254  | 0.7454  | 0.3994   | 0.2853 | 0.7291  | 0.3662    |
> | PaAno (InstanceNorm)    | **0.4307**  | **0.7928**  | **0.4141**   | **0.3830** | **0.7613**  | **0.4299**    |
>
> Long-range contextual anomalies and non-local anomalies refer to deviations defined over broad temporal structures [3,4]. PaAno does not explicitly model long-range temporal dependencies and is therefore limited in capturing anomalies that rely on extended context. Nevertheless, PaAno can detect certain non-local anomalies by aggregating the embeddings of all patches that include each timestamp, thereby capturing the local context provided by overlapping patches. PaAno can also incorporate longer temporal coverage through larger patch sizes, and our ablations show that performance remains robust—or even improves—with larger patches (see below W4!). While long-range information may matter in some cases, our empirical results on TSB-AD-U and TSB-AD-M indicate that fine-grained local patterns are often more decisive for anomaly detection.
>
> **W4. More ablation to patch length**
>
> We conducted additional ablations on patch size and found that PaAno is robust to variations in this setting. PaAno achieves strong performance at a patch size of 32 and shows slightly improved performance from 64 onward.
>
>  **TSB-AD-U**
>
> |Patch size          | VUS-PR | VUS-ROC | Range-F1 | AUC-PR | AUC-ROC | Point-F1 |
> |-------------------|--------|---------|----------|--------|---------|-----------|
> | 32 | 0.4294 | 0.8543  | 0.4113  | 0.3694 | 0.8114  | 0.4293 |
> | 64 | 0.5188  | 0.8863  | 0.4819   | 0.4559 | 0.8573  | 0.5068 |
> | 96 | 0.5291 | 0.8918  | 0.4889   | 0.4754 | 0.8697  | 0.5220    |
> | 128 | 0.5362 | 0.8862  | 0.4934   | 0.4702 | 0.8676  | 0.5191    |
>
>  **TSB-AD-M**
> |Patch size          | VUS-PR | VUS-ROC | Range-F1 | AUC-PR | AUC-ROC | Point-F1 |
> |-------------------|--------|---------|----------|--------|---------|-----------|
> | 32 | 0.3534 | 0.7582  | 0.3290  | 0.3084 | 0.7083  | 0.3580    |
> | 64 | 0.4102| 0.7864 | 0.3891   | 0.3612 | 0.7470  | 0.4071    |
> | 96 | 0.4307  | 0.7928  | 0.4141  | 0.3830 | 0.7613  | 0.4299    |
> | 128 | 0.4074| 0.7802 | 0.4031  | 0.3674 | 0.7563  | 0.4183    |
>
>
> [1] Kim et al. Reversible Instance Normalization for Accurate Time‑Series Forecasting against Distribution Shift, ICLR 2022.
>
> [2] Wu et al. CATCH: Channel-Aware Multivariate Time Series Anomaly Detection via Frequency Patching, ICLR 2025.
>
> [3] Darban et al. Deep Learning for Time Series Anomaly Detection: A Survey, ACM Computing Surveys 2024.
>
> [4] Boniol et al. Dive into Time-Series Anomaly Detection: A Decade Review, arXiv preprint,  2024.

---

> ### Author Response · Authors · 2025-11-17
>
> **Q2. Any Leakage Across Train/Test Boundaries?**
>
> For computing pretext losses during training, only temporally preceding patches within the training split are used as pretext pairs, and patches in the test split are never accessed. We confirm that the train–test boundary is strictly preserved and that no leakage from the test split occurs across windows.
>
> **W4&Q3. Ablation on alternative metric losses and rationale for choosing triplet loss**
>
> InfoNCE loss treats a large number of in-batch samples as negatives. For time-series patches, which lack absolute semantic labels, this means many negative relations are only weakly defined. But when negative samples are not clearly specified, contrastive objectives can introduce unnecessary dispersion or instability in the embedding space [5].
> To avoid this ambiguity, we assign a single farthest negative per anchor, because the farthest negative can provide highly informative contrast for refining ambiguous regions of the representation space. Triplet loss with this farthest negative produces a more stable and task-aligned metric-learning, and PaAno with triplet loss achieved consistently better anomaly-detection performance than the InfoNCE variant on both TSB-AD-U and TSB-AD-M.
>
> **TSB-AD-U**
>
> | Setting           | VUS-PR | VUS-ROC | Range-F1 | AUC-PR | AUC-ROC | Point-F1 |
> |-------------------|--------|---------|----------|--------|---------|-----------|
> | PaAno (Triplet)    | **0.5188** | **0.8863** | **0.4819** | **0.4559**| **0.8573**| **0.5068**    |
> | PaAno (InfoNCE)   | 0.4835 | 0.8670  | 0.4528   | 0.4264 | 0.8362  | 0.4775    |
>
> **TSB-AD-M**
>
> | Setting           | VUS-PR | VUS-ROC | Range-F1 | AUC-PR | AUC-ROC | Point-F1 |
> |-------------------|--------|---------|----------|--------|---------|-----------|
> | PaAno (Triplet)    | **0.4307** | **0.7928** | **0.4141** |	**0.3830** | **0.7613** | **0.4299** |
> | PaAno (InfoNCE)   | 0.3704 | 0.7619  | 0.3539   | 0.3281 | 0.7284  | 0.3721    |
>
> **Q5. Method for updating the memory bank online**
>
> PaAno can support online updates by maintaining the memory bank as a simple queue: new normal patch embeddings are added and older ones are removed, allowing the memory bank to adapt to evolving normal patterns without any re-training. We reflected this point in the revised manuscript! (see **4.2 Results and Discussion**).
>
>
> [5] Wang et al. SNCSE: Contrastive Learning for Unsupervised Sentence Embedding with Soft Negative Samples. ICIC 2023.

---

> ### Author Response · Authors · 2025-11-26
> **Waiting for response**
>
> Dear Reviewer 2Foy,
>
> As the rebuttal period is coming to a close, we would like to kindly ask whether our responses have adequately addressed your primary concerns. If so, we would be sincerely grateful if you could consider updating your score accordingly.
>
> We truly appreciate you taking the time to review our work and provide thoughtful feedback, especially amidst your busy schedule. If you have any additional questions or suggestions, we would be glad to discuss them further and incorporate any necessary improvements into the manuscript.
>
> Thank you once again for your valuable time and constructive comments.
>
> Kind regards,
>
> The Authors

---

### Official Review · Reviewer_AfnK · 2025-11-01

**Soundness:** 2
**Presentation:** 3
**Contribution:** 1
**Rating:** 4
**Confidence:** 4

**Summary:**

This paper proposes a patch-based temporal anomaly detection method that integrates memory mechanisms and representation learning to accomplish the task of temporal anomaly detection. The introduction and motivation are relatively clear and well-defined.

**Strengths:**

1. The motivation and ideas of the article are very clear, and the introduction is relatively clear;

2. The article uses more and broader evaluation metrics, rather than flawed point adjustment metrics;

3. Time series anomaly detection is a field worth exploring and has certain practical application value;

4. The article's figures are well-made, clearly presenting the content intended to be conveyed.

**Weaknesses:**

1. The novelty of the paper is limited. There has been much discussion about the Patch mechanism and Patch sequentiality in temporal tasks and temporal anomaly detection tasks, and the introduction of a memory mechanism in TSAD is not a novel contribution. The statements at the beginning of the paper are insufficient to demonstrate the novelty of the proposed framework.

2. The comparative baselines in the paper are weak. Targeting ICLR 2026, it seems that many recent SOTA methods are missing from the comparisons, especially those from 2025, which are few. It is recommended that the authors include more strong baselines to substantiate the effectiveness of their method.

3. The claimed contribution of the method in terms of lightweight design is questionable, as the results on TSB-AD show that the proposed method is slower in terms of time consumption compared to many existing methods, making it difficult to demonstrate an advantage in training and inference.

**Questions:**

1. The author needs to clearly and repeatedly clarify their contributions, especially regarding the motivation for integrating previous technologies, and whether there is truly an element of lightness, which requires stronger theoretical and experimental evidence; the presentation also needs to be adjusted.

2. It is recommended that the author supplement with more comparative work from the past 25 years, as the current comparison algorithms are insufficient.

---

> ### Author Response · Authors · 2025-11-17
>
> We would like to sincerely thank Reviewer AfnK for the insightful comments.
>
> **W1: Novelty of PaAno**
>
> Time-series anomaly detection(TSAD) methods are commonly grouped into forecasting, reconstruction, statistical, and representation based approaches. As noted in the 2024 survey by Darban et al. [1], **representation-based TSAD remains comparatively underexplored**, as only a small subset of recent representation-learning studies in the broader time-series literature explicitly address anomaly detection.
>
> PaAno advances this underexplored representation-based TSAD by introducing a patch-level embedding space specifically tailored for time-series anomaly detection. The model learns representations that are invariant to small temporal shifts yet highly sensitive to meaningful pattern differences. This is achieved by pairing each patch with slightly shifted temporal neighbors as the positives, selecting the farthest non-positive patch as the negative, and applying patch-wise normalization to handle non-stationarity. We expect these mechanisms to produce well-organized meaningful clusters of normal patches, with unseen future anomalous patches lying far from them and thus being effectively identified.
>
> Although we build upon established components drawn from diverse domains, their unified configuration here constitutes, to the best of our knowledge, the first framework explicitly designed to construct a stable and discriminative patch-representation space for time-series anomaly detection.
>
> **W2&Q2:  Comparison with recent SOTA methods**
>
> Our experiment originally included 12 recent methods published within the past three years (2023-2025). We truly appreciate your comment, which highlights the need to strengthen our comparative experiments. Therefore, we are incorporating two more SOTA methods, KAN-AD (ICML 2025) [2] and DADA (ICLR 2025) [3]. We will update the results and discussions once the new analyses are completed.
>
> **W3&Q1: Lightweightness of PaAno compared with baselines**
>
> PaAno is not the fastest model in this paper. Statistical baselines run within sub-second to 1–2 seconds, whereas PaAno requires 6.8 seconds. We acknowledge this gap and will clarify that PaAno is not intended to outperform classical models purely in speed, but to offer a strong balance between runtime and detection accuracy. In practice, PaAno delivers the highest univariate performance (VUS-PR 0.51) while remaining only a few seconds slower than the fastest neural baselines.
> In the multivariate benchmark, PaAno handles higher-dimensional inputs efficiently: its runtime (12.6 s) becomes lower than linear baselines such as DLinear (14.8 s) and NLinear (15.0 s), despite their substantially lower accuracy (VUS-PR 0.29). PaAno is also faster than many recent neural and Transformer-based methods, including PatchTST (66.9 s), CATCH (40.1 s), iTransformer (24.4 s), TimesNet (52.1 s), and OFA (532.9 s). Thus, while PaAno is not the fastest, it maintains competitive runtime and high accuracy, and its relative efficiency becomes more evident in the multivariate case.
>
> | Method Name   | Year | Type         | Run Time (Uni) | Run Time (Mul) | VUS-PR (Uni) | VUS-PR (Mul) |
> |---------------|------|--------------|----------------|----------------|--------------|--------------|
> | DLinear       | 2023 | Stat&ML      | 2.9s           | 14.8s          | 0.25         | 0.29         |
> | NLinear       | 2023 | Stat&ML      | 5.8s           | 15.0s          | 0.23         | 0.29         |
> | FITS          | 2023 | NN           | 3.1s           | 16.7s          | 0.26         | 0.21         |
> | Lag-Llama     | 2023 | Transformer  | 1220.8s        | –              | 0.27         | –            |
> | PatchTST      | 2023 | Transformer  | 26.3s          | 66.9s          | 0.26         | 0.28         |
> | OFA           | 2023 | Transformer  | 171.1s         | 532.9s         | 0.24         | 0.21         |
> | DCdetector    | 2023 | Transformer  | 5.8s           | 15.0s          | 0.09         | 0.10         |
> | TimesFM       | 2024 | Transformer  | 83.8s          | –              | 0.30         | –            |
> | MOMENT (FT)   | 2024 | Transformer  | 43.6s          | –              | 0.39         | –            |
> | MOMENT (ZS)   | 2024 | Transformer  | 42.9s          | –              | 0.38         | –            |
> | iTransformer  | 2024 | Transformer  | 9.8s           | 24.4s          | 0.22         | 0.29         |
> | CATCH         | 2025 | Transformer  | –              | 40.1s          | –            | 0.30         |
> | **PaAno (Ours)** | 2026 | NN         | **6.8s**         | **12.6s**        | **0.51**       | **0.41**       |
>
>
> [1] Darban et al. Deep Learning for Time Series Anomaly Detection: A Survey, ACM Computing Surveys 2024
>
> [2] Zhou et al. KAN-AD: Time Series Anomaly Detection with Kolmogorov–Arnold Networks, ICML 2025
>
> [3] Shentu et al. Towards a General Time Series Anomaly Detector with Adaptive Bottlenecks and Dual Adversarial Decoders, ICLR 2025

---

> ### Author Response · Authors · 2025-11-26
> **Waiting for response**
>
> Dear Reviewer  AfnK,
>
> As the rebuttal period is coming to a close, we would like to kindly ask whether our responses have adequately addressed your primary concerns. If so, we would be sincerely grateful if you could consider updating your score accordingly.
>
> We truly appreciate you taking the time to review our work and provide thoughtful feedback, especially amidst your busy schedule. If you have any additional questions or suggestions, we would be glad to discuss them further and incorporate any necessary improvements into the manuscript.
>
> Thank you once again for your valuable time and constructive comments.
>
>
>
>
>
> Kind regards,
>
>
> The Authors

---

> ### Author Response · Authors · 2025-11-29
> **Expand Experiments on 2025 SOTA methods**
>
> Dear Reviewer AfnK,
>
> As noted earlier, we completed additional experiments on **two state-of-the-art recent 2025 time-series anomaly detection methods**, KAN-AD [1] and DADA [2]. Both methods showed competitive anomaly detection performance on TSB-AD, with KAN-AD ranking second among all baselines in our evaluation.
>
> Across all experiments, including these newly added methods, **PaAno ranked first, outperforming the full set of baselines.** We sincerely appreciate your suggestion to include more recent approaches. This update meaningfully strengthened the experimental validation of PaAno. All updated results have been reflected in the final revised manuscript.
>
> [1] Zhou et al. KAN-AD: Time Series Anomaly Detection with Kolmogorov–Arnold Networks, ICML 2025
>
> [2] Shentu et al. Towards a General Time Series Anomaly Detector with Adaptive Bottlenecks and Dual Adversarial Decoders, ICLR 2025

---

### Official Review · Reviewer_6WeB · 2025-11-01

**Soundness:** 2
**Presentation:** 3
**Contribution:** 2
**Rating:** 4
**Confidence:** 3

**Summary:**

This paper presents PaAno, a lightweight and effective method for semi-supervised (normal-only) time-series anomaly detection. The authors argue that recent large-scale models (e.g., Transformers) provide diminishing returns for high computational costs, often failing to outperform simpler methods under rigorous evaluation protocols. PaAno challenges this trend by proposing a compact 1D-CNN architecture based on patch-based representation learning.

The model is trained on normal data by extracting short temporal patches. A patch encoder (1D-CNN) is optimized using a dual loss function:

Triplet Loss: A metric learning objective that clusters temporally similar patches in the embedding space.

Pretext Loss: A self-supervised classification task that predicts whether two patches are temporally consecutive.

After training, embeddings of normal patches are stored in a compressed memory bank. During inference, the anomaly score of a new patch is computed as its distance to the nearest neighbors in this memory bank, effectively measuring its dissimilarity from all learned normal patterns. Experiments on the rigorous TSB-AD benchmark show PaAno achieves state-of-the-art (SOTA) performance, ranking first across all six evaluation metrics for both univariate and multivariate data, significantly outperforming heavier Transformer-based baselines.

**Strengths:**

1. The method is architecturally simple, utilizing a lightweight 1D-CNN (0.3M parameters)  and achieving a fast runtime (Tables 2 & 3). This makes it a practical solution for resource-constrained environments, which is a commendable engineering goal.
2. The authors have conducted a comprehensive evaluation on the TSB-AD benchmark, adhering to its rigorous protocols. The inclusion of a proper ablation study (Table 4) and hyperparameter sensitivity analysis (Fig. 4) meets the requirements of a solid experimental paper.

**Weaknesses:**

Significant Lack of Novelty: This is the primary flaw of the paper. The proposed method is highly incremental and appears to be a straightforward combination of well-established, existing techniques.

- The core idea of "patch-based representation learning" for anomaly detection is directly borrowed from the computer vision domain.

- The use of 1D-CNNs for time-series feature extraction is standard.

- The use of Triplet Loss and self-supervised pretext tasks  are both common, off-the-shelf methods for representation learning.

- The paper fails to demonstrate a novel conceptual contribution. It reads more like an application of a known (visual) anomaly detection recipe to the time-series domain, rather than a new method developed from first principles.

Besides, the paper's main justification is that "local patterns matter" and that large Transformer models are "inefficient". These are not new insights. The paper fails to provide a deep analysis of why this specific combination of old techniques so dramatically outperforms other methods (including simpler ones like (Sub)-PCA ) on this benchmark. The impressive empirical result lacks a correspondingly strong conceptual or theoretical justification.

**Questions:**

The paper's justification is that "local patterns matter,"  but this is not a new insight. Simple baselines like (Sub)-PCA also operate on local windows yet perform significantly worse (Table 2). Conversely, large Transformers are fully capable of learning local patterns but also fail. This implies the success is not just about "being local." What specific properties does the 1D-CNN encoder learn from the patches—as a result of this specific dual-loss training—that (Sub)-PCA fails to capture and that Transformer-based models apparently miss?

The paper combines Triplet Loss and a Pretext Loss, both of which are well-established techniques . The ablation study (Table 4) shows both contribute. However, what is the specific synergistic effect between these two? For instance, does the pretext task (predicting temporal consecutiveness) primarily help structure the embedding space for the triplet loss to find more meaningful negatives? Or are their contributions merely additive? How critical is this specific pretext task, or would any generic self-supervised task have achieved a similar outcome?

---

> ### Author Response · Authors · 2025-11-17
>
> We would like to sincerely thank Reviewer 6WeB for the insightful comments!
>
> **W1~W4: Novelty of PaAno**
>
> Time-series anomaly detection(TSAD) methods are commonly grouped into forecasting, reconstruction, statistical, and representation based approaches. As noted in the 2024 survey by Darban et al. [1], **representation-based TSAD remains comparatively underexplored**, as only a small subset of recent representation-learning studies in the broader time-series literature explicitly address anomaly detection.
>
> PaAno advances this underexplored representation-based TSAD by introducing a patch-level embedding space specifically tailored for time-series anomaly detection. The model learns representations that are invariant to small temporal shifts yet highly sensitive to meaningful pattern differences. This is achieved by pairing each patch with slightly shifted temporal neighbors as the positives, selecting the farthest non-positive patch as the negative, and applying patch-wise normalization to handle non-stationarity. We expect these  mechanisms to produce well-organized meaningful clusters of normal patches, with unseen future
> anomalous patches lying far from them and thus being effectively identified.
>
> Although we build upon established components drawn from diverse domains, their unified configuration here constitutes, to the best of our knowledge, the first framework explicitly designed to construct a stable and discriminative patch-representation space for time-series anomaly detection.
>
> **W5&Q1: Conceptual Justification Beyond Locality**
>
> PaAno succeeds not only because it leverages locality, but also because of its embedding space constructed for time-series anomaly detection. Classical local-window methods (Sub-PCA, Sub-KNN) rely on distances or densities in the raw subsequence space, where small shifts or amplitude fluctuations can obscure structural differences crucial for anomaly detection.
>
> In contrast, PaAno detects anomalies in the embedding space that consists of well-separated coherent normal patch clusters. Because this geometry preserves the meaningful intrinsic variation of normal dynamics, an anomalous patch is far less likely to align with any of these normal clusters, enabling more reliable detection than other locality-based comparisons in the raw subsequence space.
>
> Transformer-based baselines degrade for different reasons. Prior studies show that self-attention becomes increasingly diffuse or saturated on long sequences, causing global interactions to dominate and reducing sensitivity to short-range temporal deviations [2,3,4]. Such property can degrade performance in anomaly detection, where small, localized irregularities are often decisive. By focusing representation learning on compact patches, PaAno produces embeddings that remain more responsive to fine-grained local irregularities—capabilities that Transformer models struggle to maintain.
>
> **Q2. Synergistic effect between triplet loss and pretext loss**
>
> The pretext loss was introduced to help stabilize the encoder during the early iterations of training. Because the triplet loss depends on relative distances within each minibatch, it can produce noisy or unreliable updates when the embedding space has not yet formed. Incorporating the pretext objective during this stage improves training stability, and we observed consistent gains in performance when it is used in the early phase.
>
> However, keeping the pretext loss continuously degrades performance. Once the embedding space becomes more structured, the pretext objective begins to interfere with the triplet loss, leading to degraded performance. This motivated us to limit the pretext loss to the early part of training.
>
> **TSB-AD-U**
> | Setting        | VUS-PR | VUS-ROC | Range-F1 | AUC-PR | AUC-ROC | Point-F1 |
> |---------------|--------|---------|----------|--------|---------|----------|
> | pretext 10%   | 0.5188 | 0.8863  | 0.4819 | 0.4559 | 0.8573 |	0.5068 |
> | pretext 50%   | 0.4864 | 0.8749  | 0.4661   | 0.4220 | 0.8452  | 0.4700   |
> | pretext 100%  | 0.4683 | 0.8712  | 0.4531   | 0.4043 | 0.8401  | 0.4582   |
>
>
> **TSB-AD-M**
> | Setting           | VUS-PR | VUS-ROC | Range-F1 | AUC-PR | AUC-ROC | Point-F1 |
> |-------------------|--------|---------|----------|--------|---------|-----------|
> | pretext 10%   | 0.4307 | 0.7928 | 0.4141 | 0.3830 | 0.7613 |	0.4299 |
> | pretext 50%   | 0.4010 | 0.7791  | 0.3980   | 0.3582 | 0.7481  | 0.4100   |
> | pretext 100%  | 0.3997 | 0.7662  | 0.3963   | 0.3541 | 0.7312  | 0.3998   |
>
>
>
> [1] Darban et al. Deep Learning for Time Series Anomaly Detection: A Survey, ACM Computing Surveys 2024.
>
> [2] Li et al. Enhancing the Locality and Breaking the Memory Bottleneck of Transformer on Time Series Forecasting. NeurIPS 2019.
>
> [3] Tay, Y. et al. Long Range Arena: A Benchmark for Efficient Transformers. ICLR 2021.
>
> [4] Liu et al. Non-stationary Transformers: Exploring the Stationarity in Time Series Forecasting. NeurIPS 2022.

---

> ### Author Response · Authors · 2025-11-26
> **Waiting for response**
>
> Dear Reviewer 6WeB,
>
> As the rebuttal period is coming to a close, we would like to kindly ask whether our responses have adequately addressed your primary concerns. If so, we would be sincerely grateful if you could consider updating your score accordingly.
>
> We truly appreciate you taking the time to review our work and provide thoughtful feedback, especially amidst your busy schedule. If you have any additional questions or suggestions, we would be glad to discuss them further and incorporate any necessary improvements into the manuscript.
>
> Thank you once again for your valuable time and constructive comments.
>
>
>
>
>
> Kind regards,
>
>
> The Authors

---

### Author Response · Authors · 2025-11-17

We sincerely thank all reviewers for their thoughtful and constructive feedback on PaAno. Your comments have helped us clarify the motivation, refine the details and explanations, and strengthen the contributions of this work.

We have now uploaded the **updated manuscript** and **source code** (as Supplementary Material), and all revisions are highlighted in **blue** in the revised manuscript.

**Major Changes Included in the Updated Manuscript**

- Thanks to **Reviewer 2Foy**, PaAno now adopts **Instance Normalization** to better handle regime shifts and non-stationary normal patterns, resulting in consistent performance improvements across TSB-AD-U/M.

- Thanks to **all reviewers**, we have added **comprehensive sensitivity analyses** about:

  - Model architecture (w/ height/width variations)
  - Triplet loss vs InfoNCE loss
  - Pretext loss scenarios
  - λ
  - Patch size
  - Minibatch size

- Thanks to **Reviewer 6WeB and AfnK**, we have clarified the **novelty of PaAno** in the main paper.

- Thanks to **Reviewer 2Foy and EcX6**, we have strengthened the **overall experimental discussions** and included **pseudo code of Anomaly Detection**.

-  Also, we removed the unnecessary computation steps of the pretext task after it decays to zero, making **PaAno much faster!** (TSB-AD-U: **6.9s**, TSB-AD-M: **12.8s**).

**Major Changes We Are Preparing to Include in the Updated Manuscript**

- Expanding the experiments to include two additional baseline methods published in 2025: KAN-AD [1] (ICML 2025) and DADA [2] (ICLR 2025).

[1] Zhou et al. KAN-AD: Time Series Anomaly Detection with Kolmogorov–Arnold Networks, ICML 2025

[2] Shentu et al. Towards a General Time Series Anomaly Detector with Adaptive Bottlenecks and Dual Adversarial Decoders, ICLR 2025

---

> ### Author Response · Authors · 2025-11-29
> **Final Manuscript and Source Code have been uploaded**
>
> We have now uploaded **the final manuscript** and **source code** (as Supplementary Material), and all revisions are highlighted in blue in the final revised manuscript. Please see the latest comment of authors!

---

### Author Response · Authors · 2025-11-29
**Final Revision Comments**

We would like to express our sincere gratitude to all reviewers for the time, effort, and expertise you devoted to evaluating our work. All thoughtful and constructive comments were invaluable throughout the revision process.

We have now uploaded **the final manuscript** and **source code** (as Supplementary Material), and all revisions are highlighted in blue in the final revised manuscript.

**Summary of Major Changes for the Final Revision**

- Thanks to **Reviewer 2Foy**, PaAno now adopts **Instance Normalization** to better handle regime shifts and non-stationary normal patterns, resulting in consistent performance improvements across TSB-AD-U/M.

- Thanks to **all reviewers**, we have added **comprehensive sensitivity analyses** to more thoroughly examine the design choices in PaAno. The detailed results are summarized in the following tables of final manuscript:

  - Model architecture with height/width variations (**Table 7**)
  - Triplet loss vs InfoNCE loss (**Table 8**)
  - Negative Selection in Triplet loss (**Table 8**)
  - Applying Ratio and linear decay schedule on Pretext loss (**Table 9**)
  - λ (**Table 10**)
  - Patch size (**Table 13**)
  - Minibatch size (**Table 14**)

- Thanks to **Reviewer EcX6**, we adopt **the farthest negative selection** for triplet loss, which further strengthened the performance of PaAno as well as theoretical ground of the generalization ability of PaAno to unseen anomalies.

- Thanks to **Reviewer EcX6**, we included Figure (**Figure 4**) and pseudo code (**Appendix A**) of anomaly detection procedure in PaAno.

- Thanks to **Reviewer AfnK**,  we expanded our experiments to include two recent state-of-the-art time-series anomaly detection methods published in 2025! (**KAN-AD** [1] (ICML 2025) and **DADA** [2] (ICLR 2025).)

- Also, we removed the unnecessary computation steps of the pretext task after it decays to zero, making PaAno much faster! (TSB-AD-U: 6.9s, TSB-AD-M: 12.8s).

- We improved the clarity and strengthened the explanations of the underlying theoretical mechanisms in PaAno.


**We sincerely thank you once again for the time and thoughtful attention you devoted to reviewing our work!** It has truly been an honor to receive your detailed feedback. Your comments played a central role in shaping the final version of PaAno—prompting substantial improvements, deeper analyses, and clearer explanations throughout the paper. We are deeply grateful to all reviewers.

Best regards,

Authors



[1] Zhou et al. KAN-AD: Time Series Anomaly Detection with Kolmogorov–Arnold Networks, ICML 2025

[2] Shentu et al. Towards a General Time Series Anomaly Detector with Adaptive Bottlenecks and Dual Adversarial Decoders, ICLR 2025

---

> ### Author Response · Authors · 2025-12-03
> **Final Remarks**
>
> All revisions have now been fully completed. We sincerely appreciate the reviewers once again for their time and attention throughout the process.
>
> Best regards,
>
> Authors

---

### Meta-Review · Area_Chair_AhDz · 2025-12-28

**Summary:**

The paper proposal a method for anomaly detection on time series, based on temporal convolutions to embed overlapping patches, together with multiple other clever design choices, including triplet loss, pretext task, and anomaly scoring using a memory bank of normal patches.

The paper's methodology is a combination of clever design choices, optimized for maximal benchmark performance. The results, attained on the TSB-AD benchmark (NeurIPS, 24), are strong, and substantially better than baselines, including multiple strong transformer based baselines.

The rebuttal and revisions significantly strengthened the experimental section. The authors added extensive ablations and sensitivity analyses, clarified the role of the pretext loss (useful early, harmful if kept), improved robustness to non-stationarity via InstanceNorm, and added recent 2025 baselines (KAN-AD, DADA). Runtime reporting and implementation details are clearer.

The paper gives a valuable experimental / practical contribution to ICLR. I recommend to accept it.

**Reviewer Concerns:**

Limited novelty (combination of existing methodology)
No/weak theoretical grounding
Initial gaps in baselines and experimental analysis (largely addressed)

**Reviewer Scores:**

EcX6: likely to increase from 6 to 8.

2Foy: likely remains 6.

AfnK: possibly increases from 4 to 6 after added baselines and runtime.

6WeB: stay with 4 or increase to 6.

---

### Decision · Program_Chairs · 2026-01-26

Accept (Poster)